

# A Baltic Sea estuary as phosphorus source and sink

Jakob Walve[1], Maria Sandberg[1,3], Ulf Larsson[1], Christer Lännergren[2,4]

[1]Department of Ecology, Environment and Plant Sciences, Stockholm University, SE-106 91 Stockholm, Sweden
[2]Stockholm Vatten AB, SE-106 36 Stockholm, Sweden
[3]Present adress: Stora vägen 16, SE-795 70 Vikarbyn, Sweden
[4]Present adress: Träskobacken 10, SE-129 42 Hägersten, Sweden

*Correspondence to*: Jakob Walve (jakob.walve@su.se)

**Abstract.** Internal phosphorus (P) loading from sediments, controlled by hypoxia, is often assumed to hamper the recovery
of lakes and coastal areas from eutrophic conditions. We use a box-model to calculate seasonal and annual inputs, export, retention and internal cycling of P in the inner archipelago of Stockholm, Sweden (Baltic Sea) in 1968–2015. The area receives freshwater from Lake Mälaren and treated sewage from the greater Stockholm area. The sewage treatment plants (STPs) have improved their nutrient removal in steps, starting with P in 1972 and nitrogen in 1996. In the first 10-20 years after the main P load reduction in 1972-76, the model shows, in comparison to the load, a small negative annual P balance,
probably due to release from legacy sediment P stores. The now stabilized, near neutral P balance indicates no continued internal loading from legacy P, but P retention is low, despite improved oxygen conditions. Seasonally, sediments are a P sink in spring and a P source in summer and autumn. Most of the deep-water P release from sediments in summer-autumn appears to be derived from the settled spring bloom and is exported during winter. Oxygen consumption and P release in the deep water are generally tightly coupled, indicating limited control by P binding to iron-oxyhydroxides under oxic
conditions. However, in years of deep-water hypoxia enhanced P release suggest contribution from redox-sensitive P stores. The oxygen conditions in the area have generally improved, probably due both to lower sedimentation of organic matter from the 1970s and lower STP ammonium loads from the late 1990s. Increased oxygen inputs to the intermediate and deep waters due to weakened stratification and enhanced vertical mixing have probably also contributed, while increased respiration rates due to elevated bottom water temperatures probably explain worsened oxygen conditions during the 1990s.
Since the P turnover time is short and legacy P minute, measures to bind P in Stockholm inner archipelago sediments would primarily accumulate P imported from the Baltic Sea and from Lake Mälaren inflow, and management here should focus on reducing external nutrient inputs.

## 1 Introduction

Bottom water hypoxia (here oxygen concentration <2 mg L$^{-1}$) and anoxia is a common eutrophication problem in stratified
lakes and coastal waters (Diaz and Rosenberg, 2008; Conley et al., 2011). Its root cause is excessive inputs of nutrients resulting in increased settling of organic matter, which increases oxygen (O$_2$) consumption. Variations in deep-water ventilation also influence the incidence of hypoxia (Zhang et al., 2010). In addition to its direct adverse effects on benthic



fauna, hypoxia also influences nutrient cycling processes, with feedback effects on water quality. It is well known that when bottom waters turn anoxic, phosphorus (P) release from sediments can be drastically enhanced (e.g. Einsele, 1936; Mortimer, 1941; Middelburg and Levin, 2009). Such "internal P loading" is often assumed to hamper the recovery of eutrophicated ecosystems (Vahtera et al., 2007; Schindler, 2006, 2012; Stigebrandt et al., 2014; Puttonen et al., 2014).

In the Baltic Sea, one of the world's largest brackish ecosystems, control of both external and internal P loading is considered crucial for reducing the recurring summer blooms of nitrogen-fixing cyanobacteria (Vahtera et al., 2007). Here, large fluctuations in water mass P inventory are linked to the areal extent of hypoxia (Conley et al., 2002; Jilbert et al., 2011) and numerous experimental studies have documented drastically increased sediment P-release as formerly oxic sediments turn anoxic (e.g. Koop et al., 1990; Gunnars and Blomqvist, 1997). The mechanism usually invoked is the classical redox-
sensitive release of dissolved inorganic P (DIP) from P bound to iron(III)oxyhydroxides (Fe-P) as Fe is reduced to Fe(II) (Mortimer, 1941; Mort et al., 2010). The importance of $O_2$ control of P release from sediments is also supported by in-situ flux studies showing that oxygenation of anoxic sediments can drastically decrease the P release (Ekeroth et al., 2016a; Hall et al., 2017).

      The long-standing paradigm that "$O_2$ controls P release from sediments" is however too simple to fully describe the
sediment P release processes (Gächter and Müller, 2003; Hupfer and Lewandowski, 2008). On a longer time-scale the P release from sediments is the imbalance between the P sedimentation and the P binding capacity in the deeper, usually anoxic, sediment (Hupfer and Lewandowski, 2008; Carey and Rydin, 2011; Rydin et al., 2011). Even though Fe(III)-bound P can be abundant in surface sediments in the Baltic Sea, it seems to contribute little to the long-term P retention, since P is permanently buried mainly in organic forms (Jensen et al., 1995; Lukkari et al., 2009; Rydin et al., 2011; Mort et al., 2010)
and, under some conditions, as persistent Fe(II)-P minerals (Slomp et al. 2013; Reed et al. 2016). In marine sediments sulphate reduction promotes the binding of Fe as sulphides, decreasing the Fe available for P-binding, and allowing considerable P release from oxic surface sediments (Caraco et al., 1990; Blomqvist et al., 2004). This suggests that the sediment P release is tightly coupled to sedimentation and organic matter decomposition (Caraco et al., 1990) and that lower $O_2$ concentrations are often the result of organic matter mineralization rather than the controlling factor for P release (Hupfer
and Lewandowski, 2008).

      Baltic Sea coastal archipelagos and bays are affected by P inputs with local freshwater runoff, import from the open Baltic Sea and sometimes also by direct discharges of treated municipal sewage water. Often, internal P loading from sediments is regarded as an important additional P source (e.g. Puttonen et al., 2014; Rydin et al., 2017). Moreover, it is often claimed (e.g. Schindler and Vallentyne, 2008), that following high P loading, historical P storages in coastal sediments
may leak for many years, preventing the recovery of good water quality. Indeed, a high load of organic matter can increase the sediment organic P fraction (Lukkari et al., 2009), which potentially contributes to internal loading as external loading is relieved. Accumulated organic matter will also elevate oxygen demand, delaying improvement of $O_2$ conditions (Middelburg and Levin, 2009; Reed et al., 2011). In lakes, recurrent seasonal internal P loading can be an important factor delaying

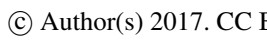



ecosystem recovery (Schindler, 2006; Nürnberg and LaZerte, 2016). In contrast, coastal estuaries usually have high water-turnover rates and the resulting larger export of released P should facilitate water quality recovery.

The Stockholm archipelago, NW Baltic Sea Proper (Fig. 1), has historically received high nutrient loading as sewage from the Stockholm area (Brattberg, 1986) and with freshwater from Lake Mälaren (at LM in Fig. 1), which drains a large

inland area with agriculture, forests and small to medium towns. Due to increasing discharges of mostly untreated sewage water, water quality in the archipelago had deteriorated badly by the mid-20th century (Brattberg, 1986; Schindler and Vallentyne, 2008). In the early 1970´s, effective BOD (biological oxygen demand) reduction and chemical P precipitation was implemented in Stockholm's major sewage treatment plants (STPs) and around Lake Mälaren (Karlgren and Ljungström, 1975). Archipelago water quality then improved rapidly, within a few years P concentrations and phytoplankton

biomass, particularly of cyanobacteria, decreased substantially (Anon, 1975; Karlgren and Ljungström 1975; Brattberg 1986; Johansson and Wallström, 2001). Following Sweden's accession to the European Union in 1995, the two largest STPs discharging to the inner Stockholm archipelago had nitrogen (N) removal installed from 1996, which also slightly improved P removal. As a result, water quality improved further, both in the still mostly P-limited inner archipelago and further out, where N now decreased to limiting levels (Boesch et al., 2006; Lännergren and Stehn, 2011).

Over 40 years have passed since the P and BOD loads to the Stockholm inner archipelago were drastically reduced. Levels of P and chlorophyll have declined and oxygen conditions have improved in the inner archipelago (Boesch et al., 2006; Norkko et al., 2011). Still, the sediments are suggested to be an important P source, judging from the late summer increase of P concentrations below the summer pycnocline that propagate to the surface waters in late summer to autumn (Boesch et al., 2006). However, it is unclear how much of this is a seasonal recycling of P due to organic matter

decomposition, the extent to which the P release is controlled by low $O_2$ concentrations, and whether historical legacy P is still influencing the area.

While numerous studies have dealt with the short-term $O_2$-dependent release of dissolved inorganic P (DIP) from Baltic Sea coastal sediments (e.g. Ekeroth et al., 2016b), fewer studies focus on the long-term retention of P, relative to external P inputs (e.g. Almroth-Rosell et al., 2016). Even fewer studies explore the dual role of sediments as sink and source

for P also over seasonal and annual scales. This may partly be a methodological issue. Although direct in-situ measurements of sediment release or uptake of DIP are relatively straightforward (e.g. Hall et al., 2017; Sommer et al., 2017), they require sophisticated equipment and a very large sampling effort to representatively cover different depths, sediment types and seasons. Moreover, these methods do not quantify the settling fluxes of P to the sediments and thus are not sufficient to determine if the sediments is a net P sink or source (Nielsen et al., 2001; Hupfer and Lewandowski, 2008). Unfortunately,

direct sedimentation measurements are prone to large uncertainties (Blomqvist and Larsson, 1994; Gustafsson et al., 2013).

Box models can help constrain and integrate net nutrient fluxes over larger coastal areas, for full seasonal cycles and for longer time-periods (e.g. Savchuk, 2005). In this study, we use a dynamic box model to analyse a time-series of nearly 50 years of water P and $O_2$ concentrations in the Stockholm inner archipelago. Previous P budgets for the inner Stockholm archipelago (Anon., 1975; Karlgren and Ljungström, 1975; Karlsson et al., 2010; Almroth-Rosell et al., 2016) have focussed



primarily on the long-term P balance. We also calculate the area's seasonal and annual function as source and sink for P relative to the $O_2$ concentrations and the $O_2$ consumption rates. We test the sensitivity of our model results by varying some basic assumptions, e.g. of the origin of saline water flowing into the inner archipelago and of the estimated errors of P concentrations, fresh water input and STP loads.

**2 Methods**

**2.1 Study site**

The Stockholm archipelago comprises ~24000 islands (Hedenstierna, 1948) (Fig. 1). Its dominant freshwater source is Lake Mälaren (drainage basin 22000 km$^2$) that empties through Norrström in central Stockholm (LM in Fig. 1) to the relatively enclosed inner archipelago (IA). This connects to the middle archipelago through one major and three shallow minor straits.

The 18 m deep Oxdjupet sound (O in Fig. 1) has the only significant deep-water inflow and ~84 % of the surface water outflow (Engqvist and Andrejev, 2003). Water exchange between the inner and middle archipelago is governed mainly by estuarine circulation, generated by the large freshwater flow, while wind and barotropic forcing are less important (Engqvist and Andrejev, 2003), making the IA suitable for mass-balance calculations. The IA local drainage area is 323 km$^2$, its water surface area 103 km$^2$, its mean depth 14 m and its maximum depth 57 m.

**2.2 Data sources**

We used water chemistry data from the monitoring program of Stockholm Vatten (1968–2015) and the investigations of Mats Waern in 1968–1976 (Nordling et al., 1984). Data from 1968–1981 were digitized from printed reports. Sampling frequency is detailed in Supplement 1. Dissolved inorganic phosphorus (DIP) and total phosphorus (TP) were analysed by standard colorimetric methods (molybdate reactive P), TP after persulphate digestion. Due to analytical problems, nutrient

data from 2011 were not used (Walve, 2012). Temperature was measured in water samplers until 1991 and afterwards at the same depths with a CTD probe. Salinity was calculated from conductivity of water samples in 1968-1991 and 2011-2015, and measured in-situ by CTD in 1992-2010. Oxygen was measured by Winkler titration (SS-EN 25813-1), hydrogen sulphide ($H_2S$) by spectrophotometer (SS 028115-1) and converted to negative oxygen. On a few occasions, surveys by Stockholm Vatten and the Mats Waern group sampled the same station on the same day, with good agreement (salinity

$r^2$=0.99, slope=0.99, n=19, TP $r^2$=0.99, slope=0.98, n=15).

Data on the daily Lake Mälaren freshwater discharge, calculated by Stockholms Hamnar AB (Ports of Stockholm), were provided by Stockholm Vatten and have an estimated error of <5 % (SMHI, 2006). Concentrations of TP and DIP in the discharge of Lake Mälaren were measured monthly (1968–1973) or weekly (from 1974) by Stockholm Vatten. Sewage treatment plant (STP) discharges of water and TP to the inner archipelago were obtained from Stockholm Vatten and

Käppalaförbundet (Henriksdal STP – quarterly data 1971–1987, weekly 1988–2013, monthly 2014-15, Käppala STP– quarterly 1973–1985, weekly 1986–2015, Loudden STP – quarterly 1971–1988, weekly 1989–2003), Bromma STP –





weekly 1989–2013, monthly 2014-15 and Hemmesta STP – weekly 2004–2006). For 1968–1970, we only found summed annual TP load from all STPs discharging into the inner archipelago (Anon., 1975). DIP load, analysed on unfiltered water with an unknown fraction of Fe-bound P, was available from 1989. The Swedish Meteorological and Hydrological Institute (SMHI) provided data on precipitation (station Stockholm 98210) and local land runoff 1977-2015 (S-HYPE model). We

used the monthly means for 1977-2015 to estimate local runoff for 1968-1976. Since local runoff is <2 % of total annual fresh water inputs, resulting errors are small.

**2.3 Water and salt box-model**

We used a box-model (Fig. 2) with four depth layers to calculate the water, salt and P budgets of the IA. The model assumes an estuarine circulation with out-flowing brackish surface water and a counter-current of in-flowing saltier water (SW) from

the middle archipelago to the deep water of the IA. Model box volumes (392 $Mm^3$ for upper surface $US$ 0-4 m, 465 $Mm^3$ for lower surface $LS$ 4-10 m, 465 $Mm^3$ for intermediate $M$ 10-20 m, and 214 $Mm^3$ for the deep water $D$ 20-57 m) were calculated from hypsographic curves of IA basins (SMHI, 2003), excluding three relatively small and isolated, peripheral basins (Brunnsviken, Edsviken and Kyrkfjärden).

  We used total freshwater inflow ($FW_t$), salinity of the IA and salinity of inflowing SW to constrain water flows, with

two different approaches. Firstly, we used volume-weighted mean salinity of the IA to constrain water flows, typical for conventional one-dimensional box models. Thus, this "*mean model*" yields a SW inflow matching the salt amount in the IA and the salinity of outflowing water (as volume-weighted mean for surface water, i.e. $Sal_{out}$ equals $Sal_{US}$, Fig. 2). We used this approach to estimate representative mean internal P fluxes in the IA. For some analyses, we used a secondary "*boundary model*". It adds an extra boundary SW inflow needed to match the salinity in the surface water near the boundary, which is

higher than the volume-weighted mean surface water salinity of the IA. Thus, the boundary model accounts for the surface water salinity gradient by adding to the water turnover in the outer part of the IA, not represented in the mean model. We give detailed calculations below. We used the boundary model to correct the P input-output budget but do not regard it as representative for the mean internal P budget, since it is mainly a "short-circuit" flow occurring in the outermost sub-basins of the IA. We made several sensitivity tests to check the effects of various assumptions on the results.

Equations for water exchange used Knudsen's theorem for estuarine circulation (Knudsen, 1900), which assumes constant water volume (Eq. 1) and equilibrium of salt transport (Eq. 2),

$$Q_{out} = Q_{in} + Q_{FWt} \qquad (1)$$

$$Q_{out}\, Sal_{out} = Q_{in}\, Sal_{in} \qquad (2)$$

where $Q_{out}$ is the out-flowing volume of surface water (with the relatively low salinity $Sal_{out} = Sal_{US}$ in the mean model),

$Q_{in}$ is the in-flowing volume of deep water (with the relatively high salinity $Sal_{in}$), and $Q_{FWt}$ is the total fresh water input to the estuary, i.e. the sum of Lake Mälaren discharge, local land runoff, precipitation and waste water discharge. $Q_{in}$ is calculated from equations (1) and (2) according to





$$Q_{in} = Q_{FWt}\,Sal_{out}/(Sal_{in} - Sal_{out}) \qquad (3)$$

Since the Knudsen equations assume steady-state, which for salt is a valid approximation only for a period longer than the turn-over time of the estuary, where volume-weighted salinity can shift with up to one unit within a month, we modified the salt mass-balance equation to include changes in total salt amount, $\Delta S_{IA}$, of the IA,

$$\Delta S_{IA} = Q_{in} Sal_{in} - Q_{out}\,Sal_{out} \qquad (4)$$

$Q_{in}$ is calculated from Eq. (1) and (4) according to

$$Q_{in} = (\Delta S_{total} + Q_{FWt} Sal_{out})/(Sal_{in} - Sal_{out}) \qquad (5)$$

This is similar to the box model approach developed by Hagy et al. (2000) and used by Testa and Kemp (2008), Testa et al. (2008) and Boynton et al. (2008) to calculate salt and water transport in the Patuxent River estuary of Chesapeake bay.

The model interpolates the daily changes in salt amount $\Delta S_{IA}$ from volume-weighted salinity data for each depth layer, calculated from measured salinity profiles (4 m depth resolution) at the central stations (A, AV, H, L, K in Fig. 1), and water volumes for 1 m depth layers (SMHI, 2003). We tested the representativeness of the central stations with data from all sub-areas 1998–2005. The central stations slightly underestimated salinity and DIP, but we found no systematic difference for TP (Suppl. 2). We used these relationships to correct data for each depth layer. A model sensitivity test with non-adjusted data

showed smaller uncertainty than for other sensitivity tests (see below). In some years absent winter (February) data for the IA central stations were estimated from Solöfjärden (S) data using correlations to the central stations, and correlations between IA surface water and deeper layers for years with full February data sets (see sampling frequency in Suppl. 1). December data were generally lacking but model sensitivity tests, run with extrapolated November or February data, rather than from linear interpolation, showed less divergence than for other sensitivity tests (see below).

Since there is often a shallow pycnocline at high fresh water input, the volume-weighted mean salinity of the 0-4 meter layer was used as the salinity of the out-flowing water. On most occasions the upper water mass is well-mixed to 10 m, suggesting a deeper out-flowing layer, but the 0–4 m layer salinity is still representative due to the weak vertical salinity and nutrients gradients on these occasions. The salinity at 20–30 m depth in Trälhavet (T in Fig. 1), just outside the boundary strait Oxdjupet (O in Fig. 1), was used to represent in-flowing deep water (Lännergren and Stehn, 2011). This salinity is in

good agreement with the bottom water salinity in Solöfjärden (S in Fig. 1), the IA sub-basin closest to Oxdjupet. Oxdjupet data also support the use of this layer (Lännergren and Stehn, 2011), but were not used in the model, due to large data gaps.

The wind-driven and diffusive vertical mixing (Fig. 2) was calculated according to the salt mass-balance equations for the bottom layer $D$ (Eq. 6), intermediate layer $M$ (Eq. 7), and the upper surface layer $US$ (Eq. 8):

$$\Delta S_D = Q_{in} Sal_{in} - Q_{in} Sal_D + Q_{Dmix} Sal_M - Q_{Dmix} Sal_D \qquad (6)$$

$$\Delta S_M = Q_{in} Sal_D - Q_{in} Sal_M + Q_{Dmix} Sal_D - Q_{Dmix} Sal_M + Q_{Mmix} Sal_{LS} - Q_{Mmix} Sal_M \qquad (7)$$

$$\Delta S_{US} = Q_{in} Sal_{LS} - Q_{out} Sal_{US} + Q_{Smix} Sal_{LS} - Q_{Smix} Sal_{US} \qquad (8)$$



where $\Delta S$ is the change in salt amount in each layer for the time step used, $Q_{Dmix}$ is the vertical mixing of water layers $D$ and $M$ (10-20m), $Q_{Mmix}$ is the mixing of layers M and $LS$, and $Q_{Smix}$ is the mixing of layers $LS$ and $US$ (Fig. 2). Mixing terms were calculated by rearranging the equations.

The boundary model adds an extra SW inflow $Q_{inxD}$ according to

$$Q_{inxD} = (Sal_b \, Q_{out} - Sal_{out} \, Q_{out})/(Sal_{in} - Sal_b) \qquad (9)$$

where $Sal_b$ is surface water salinity at the boundary. This inflow generates a corresponding increase in outflow.

The salinity profile 20–57 m of the IA central stations is relatively uniform and similar to the deep water in Solöfjärden, indicating that salinity in the deep water is determined mainly by inflowing water from the middle archipelago and that mixing with the overlying water-mass is small. The main vertical salinity gradient is generally found in the intermediate 10–20 m layer (*M*). In the innermost part of the IA, discharged sewage water creates a 4-layered water flow, with an out-flowing sewage-enriched layer at 8–20 m, below inflowing shallow and an outflowing surface layers and above inflowing deep water (Lännergren and Stehn, 2011). These intermediate layers are restricted to the innermost part of the IA and do not affect the water and salt budgets of the box-model. Moreover, directing STP flows to either US or M layers has no effect on our results as presented, since we aggregate results for the upper 0-20 m.

The modelling used the software ExtendSim8 (Imagine That Inc.), which is suitable for sequential computations and interpolations of large data sets. We used a time step of 1 day and computation $dt$ =100 day$^{-1}$.

We correlated model results with water temperature data from the representative station Koviksudde (K) with the most complete sampling. Data were compiled for depth intervals with consistent data for all sampling occasions. Means were calculated for depth intervals 0-8 meters (data mostly from 0, 4 and 8 m), 12-20 m (mostly 12, 16 and 20 m), and 24-32 m (24, 28 and 32 m). Since some sampling occasions were close to the end or the beginning of months, temperature data were interpolated between occasions before calculations of seasonal means.

**2.4 Phosphorus model**

Phosphorus inputs to the IA come from Lake Mälaren ($P_{LM}$), local land runoff ($P_{LR}$), precipitation on the sea surface ($P_P$), STPs ($P_{STP}$) and inflowing salt water from the middle archipelago ($P_{SW}$). P import ($P_{SW}$) from the middle archipelago was calculated from the volume of in-flowing water $Q_{in}$ and the mean P concentration at 20–30 m depth in Trälhavet (T in Fig. 1). Atmospheric P deposition on the IA is ~15 kg km$^{-2}$ yr$^{-1}$ (Savchuk et al., 2008) – minute compared to other P sources.

In the mean model (defined above), P export ($P_{EXP-MM}$) is a function of the mean model outflow and the volume-weighted P concentration in the upper surface layer. The P concentration in this and all other layers was continuously adjusted to fit interpolated observations, by additions or removal from the model. The summed corrections ($P_I$) were used to estimate net internal losses and inputs to the water column.

In the boundary model, P export ($P_{EXP-BM}$) was calculated using the out-flow of the mean model and the P concentration at 0–4 m depth near the boundary (defined by stations S, O and TF, Fig. 1), and then adding the extra flow across the boundary that was needed to match the boundary salinities. The TP export calculated by the boundary model also



had the advantage of larger data availability of TP concentrations at the boundary than at the central stations of the IA (Suppl. 1). Using the boundary model, monthly and annual net external P import–export balance was calculated. The extra flows across the boundary generally resulted in additional annual net P loss from the inner archipelago, due to higher P concentrations in the outflowing surface water than in the inflow (defined by concentration in deep water at station T).

## 2.5 Oxygen budget for the deep water

The $O_2$ budget was calculated only for the deep layer (>20 m). In the model, $O_2$ is imported to the deep water by SW inflow and mixing with overlying water (10-20 m). Oxygen is exported by upwelling and mixing into overlying water. Consumption of $O_2$ in the deep water was calculated as for P, i.e. with a function continuously adjusting the estimated concentration to fit observations. The $O_2$ consumption was converted to a theoretical mineralization of P assuming Redfield composition (1 mol P per 106 mol $O_2$) of organic matter. The complete oxidation of ammonium, generated from organic matter, to nitrate would give a 30 % lower P yield (1 mol P per 138 mol $O_2$). We used the higher P yield (1:106) because of observed periodical accumulation of ammonium in the deep water, unknown ammonium transfer to intermediate water layers, and unknown contribution of nitrate to organic matter oxidation (denitrification).

## 2.6 Sensitivity analysis

We tested the robustness of the P budgets by 1) varying the fresh water flow from Lake Mälaren by ±5 %, 2) applying different recruitment depths of inflowing SW (12–20, 16–24 and 24–32 m compared to the default depth 20–30 m) from the middle archipelago, 3) assuming a ±10 % error in the P discharge from the STPs, 4) assuming a ±10 % error in TP and DIP measurements, and 5) changing the depth layer of the out-flowing surface water to 0-10 meters.

## 3 Results

### 3.1 Salinity and temperature

The salinity of the deep water (>20 m) in the IA closely mirrors the salinity at 20–30 meters depth in Trälhavet (T), the assumed recruitment depth for inflowing salt water (SW) (Fig. 3a). The SW salinity peaked around 1980 at 6.0-6.5 and has since slowly declined and stabilized at ~5.5. The volume-weighted mean salinity of the IA shows considerable seasonal variation (Fig. 3a), mainly due to variable fresh-water input affecting the surface water salinity (Fig. 3b). The long-term trend of IA mean salinity mirrors that of the deep water but shows a more abrupt decline, influenced by the large FW inflows in 1977-1988 (Fig. 4a). From the late 1980´s, a less distinct halocline and a deeper surface mixed layer (Fig. 3b) accompanied the salinity decrease in the deep water. The pycnocline depth (depth of maximum density change per meter) increased from ~10 to ~14 m and its strength (density change per meter) decreased (data not shown but indicated in Fig. 3b). From 1989, there was an abrupt increase (~2°C) of mean July to October temperature in the intermediate and deep layers (Fig. 3c). There is a seasonal temperature rise from July to October (of ~3 °C), but this showed little change, and the higher



mean July-October temperature from 1989 was mostly the result of larger seasonal temperature rises earlier in the year. From 1989, early May temperatures were ~0.5°C and ~1°C higher in the intermediate and deep layers, respectively, and early July temperatures ~2°C and ~1°C higher. After 2007, temperatures appear to have increased even further (Fig. 3c).

## 3.2 Phosphorus concentrations

Following the reduction of phosphorus (P) load from the STPs in the early 1970´s (see below), yearly mean total P concentrations (TP) in the IA decreased from >100 µg L$^{-1}$ to around 49 µg L$^{-1}$ in 1976-85 and 39 µg L$^{-1}$ in 1986-1995 (Table 1, Fig. 3d, e). After the P-treatment upgrade in 1996, TP declined further to 30-32 µg L$^{-1}$ (Table 1). In 1968-75, TP was lower in the deep than in the surface water, but this was reversed after the P load reduction (in the following, P refers to TP unless otherwise stated). The P decline was less prominent in the deep water than in the surface water and the deep water varied more among years (Fig. 3d, e). In the early 1970´s, substantial amounts of dissolved inorganic phosphorus (DIP) remained in the upper 20 m all year (Fig. 3d). From the mid-1970´s, summer (June–August) DIP concentrations declined to values close to or below the analytical quantification limit (3 µg L$^{-1}$). In the deep water, DIP was consistently the main P fraction (Fig. 3e). After the P load reduction in the 1970´s, the P concentration in the inflowing salt water (SW) from the middle archipelago (Trälhavet) has declined from ~35 to ~25 µg L$^{-1}$ (Table 2).

## 3.3 Water budget

Total fresh water input (FW$_t$) to the IA 1968-2015 (5296 ±1418 Mm$^3$ yr$^{-1}$, or 168 ±45 m$^3$ s$^{-1}$, mean ±SD among years) was dominated by Lake Mälaren (94 ±2.1 %), followed by treated sewage (3.6 ±1.2 %), local runoff (1.7 ±0.6 %) and direct precipitation on the water (1.1 ±0.4 %). FW$_t$ input shows no trend, but the early 1970´s were relatively dry (1976 extremely dry), followed by a period of high discharge in the late 70´s to the late 80´s (Fig. 4a, Table 1). In the 90´s there was again relatively low discharge (1996 and 1997 were especially dry), followed by high discharges around year 2000 and at the end of the time series (Fig. 4a, Table 1). On average, the annual salt water inflow from the middle archipelago (Q$_{SW}$) was 74 ±22 % and 136 ±54 % of the FW$_t$ input, calculated by the mean and boundary models, respectively (Fig. 4b, Table 2). Salt-water inflow showed less inter-annual variation than the FW$_t$ discharge (Fig. 4), and on an annual basis, there was no correlation between these flows. The SW inflow was higher at the end of the time series, especially when calculated by the boundary model (Fig. 4b, Table 2). The sensitivity analyses with variable recruitment depth of SW, or variable discharge from Lake Mälaren, changed the SW inflow only ~6 % relative to default models, except for the scenario with shallowest (12-20 m) recruitment depth (+15 %), and the scenario with export depth 0-10 m (+36 %). The mean model gives an annual mean water turnover time of 2.2 ±0.6 months. The seasonal variation is large, ranging from 1.7 months in April to 5.2 months in August.



### 3.4 External P loads

In the late 1960's and early 1970's, the direct P discharges from STPs to the inner archipelago were very high, peaking at over 800 tons in 1970 (Fig. 5a). In 1976, when P treatment had been implemented, P loads from STPs had decreased by a factor of ten (Fig. 5a). Reduced P loads from other STPs to Lake Mälaren substantially decreased also FW (FW$_t$ minus IA

STPs) P inputs during this period. From 1976, the annual P load from STPs has mostly been considerably smaller than the P inputs from FW or SW inflow (Fig. 5b, Table 1). From 1976 to 1995, the P load from STPs decreased gradually from ~80 to 45 tons yr$^{-1}$ and after the 1996 P-treatment upgrade was reduced to ~30 tons yr$^{-1}$, i.e. approximately 10 % of the total external (STPs, FW and SW) P load to the inner archipelago. From 1976, the mean annual P loading with FW has been 170 ±62 tons yr$^{-1}$, most of which from Lake Mälaren (162 ±62 tons yr$^{-1}$, Table 1). Although there is a significant increase of the modeled

inflowing SW volume with time, there is no equivalent increase in the SW P input, because of the decreasing SW P concentrations in Trälhavet (T in Fig. 1, Table 2). Atmospheric P deposition is negligible in the P budget of the inner archipelago, contributing only about 1.5 tons yr$^{-1}$ (using 15 kg P km$^{-2}$ yr$^{-1}$ from Savchuk et al., 2008).

The Lake Mälaren and SW DIP inputs (data not shown) follow the same pattern as the TP load. From 1989 (when data on DIP loads from STPs are available) on average 15 tons of DIP per year originate from STPs, 10 % of the total

external DIP inputs to the area, or 17 % of DIP load from land. In contrast to the decrease in TP load, DIP load from STPs remained virtually unchanged after the upgraded P-treatment in 1996, indicating that mainly particulate P was removed.

### 3.5 Annual and long-term P input-export budget

The annual P export from the inner archipelago closely mirrored the summed annual external P loading (including P in SW inflow) (Fig. 5c). Thus, annual P loading mainly governed the P export. The input-export difference shows the net P balance

of the area, with positive values indicating that the area is a net P sink (export < input, i.e. positive P balance and [positive] net P retention), while negative values indicate it is a net P source (export > input, i.e. negative P balance). The annual input-export difference was generally a small fraction (mostly <10 %) of the inputs and exports and showed considerable variation among years (Fig. 5c, d). The boundary model sensitivity analysis indicated an uncertainty of yearly P balance of approximately ±25 tons (Fig. 5d). Following the major relaxation of external P loads in the 1970´s, the long-term mean P

balance was initially negative (1976-1985 and 1986-1995), but later became slightly positive (1996-2005 and 2006-2015) (Table 3). Sensitivity tests show largest influence of assumed SW recruitment depth, followed by potential errors in measured TP concentrations and in 1996-2005 and 2006-2015 mean P balance ranged from -19 to +15 tons/year, excluding the most extreme scenario with the shallowest recruitment depth (Table 3). For 1976-2015, linear regression indicates a weak long-term trend of increasing P retention (0.9 tons year$^{-1}$, r$^2$=0.16, p<0.02), for 1996-2015 there was no trend (r$^2$<0.01,

p>0.8) (Fig. 5d).

In 1968-74, before full P reduction in STPs, P retention was highly variable (Fig. 5d). As STPs in this period were the main source of P, assuming 10 % uncertainty in the STP load will mean a proportionally much larger uncertainty in the




retention value. In 1969 and 1970, years with relatively good $O_2$ conditions (Fig. 6b), a positive P retention was relatively robust. From 1976, when the P loads had decreased substantially, our results indicate a 9-year-period with mostly negative P balance, i.e. the export was larger than input. The discrepancy between the boundary model and the mean model for some of these years (Fig. 5d) was due to better data availability in winter, the season with most of the P export. This was followed by

a 5-year-period (1985-1989) with a balanced P input and export which coincided with better $O_2$ conditions both in the deep (>20 m) and intermediate (10-20m) water. As $O_2$ conditions deteriorated again in the 1990´s, there was clear net P loss from the IA. This may partly have been release of P deposited in the previous more $O_2$-rich period (1985-89). In 1992 and 1993, the high P loss from the IA (Fig. 5d) was associated with relatively wide-spread hypoxia both in the deep water and the 10-20 m layer (Fig. 6b), and exceptionally poor P retention <20 m (see next section and Fig. 6a), suggesting much P release

from shallow bottoms.

### 3.6 Internal phosphorus fluxes

Annual P budgets were also derived separately for the deep water (>20 m) and for the combined surface and intermediate layers (0-20 m), using the mean model. For each layer, we calculated internal losses or sources of P from the sum of daily adjustments of the P modelled from external inputs that were required to match the observed P. In the upper 0-20 m layer,

the net loss of P needed to balance the annual P budget was interpreted as due to net P sedimentation (Fig. 6a). Conversely, a source of P was needed in the deep-water (Fig. 6a), mostly in the form of DIP (results not shown, but see Fig. 3e), and was interpreted as a net sediment P release. The deep-water sediments released on average $67 \pm 19$ tons P $yr^{-1}$ (2.1 g $m^{-2}$ $yr^{-1}$, i.e. 5.8 mg $m^{-2}$ $day^{-1}$, or 0.19 mmol $m^{-2}$ $day^{-1}$ for 32.3 $km^2$ bottom area below 20 m depth) in 1976–2015, with no trend (Fig. 6a). The annual net P loss from 0-20 m was similar, on average $61 \pm 29$ tons for 1976–2015. The estimated deep-water P-release

was most sensitive to assumed potential errors in P concentrations and SW recruitment depth (Table 4).

Most net P sedimentation from the upper 20 m layer (~56 % of annual) occurs in spring (March-May), most of the deep-water net P release (~57 %) in summer and autumn (July-October, Fig. 6c and Fig. 7). Maximum P release usually occurs in September (13 tons per month, i.e. 13 mg $m^{-2}$ $day^{-1}$ or 0.43 mmol $m^{-2}$ $day^{-1}$). In spring, the net P loss from surface layers decreases the P pool and removes some of the external P input, and the P export is smaller than the input (Fig. 7). In

late summer, when the P pool starts to rebuild, there is net P input in the deep-water (partly also advected to surface layers) and small net P loss from surface layers (Fig. 7). Also contributing to increasing P concentrations in summer and autumn are the larger external P inputs compared to P exports. The yearly net P sedimentation in spring (March-May) correlated with deep-water sediment P release in summer and autumn (July-October) (Fig. 8a), which is consistent with a predominantly internal recycling of P within the plankton growth season March-October. Although there is on average a net P loss from the

0-20 m layer also in summer, it varies considerably among years. In some years, there is a net internal P source in the upper layer 0-20 m in summer, suggesting larger P release from shallow sediments than gross P sedimentation from the surface water.





The combined annual net release or loss for all water layers indicates if the inner archipelago sediments acts as a source or sink for P relative to the water column (Fig. 6a). The pattern was similar to the net input-export budget (Fig. 5d), i.e. in years with net P release to the water, there was often, but not always, a larger net P export. Differences are partly due to the different approaches used (mean and boundary models, respectively), but delays between years also contribute, as P
accumulated in the water late in the year is partly exported early in the following year.

### 3.7 Oxygen concentrations

Oxygen ($O_2$) concentrations in the deep water vary seasonally, being highest in January–May and lowest usually in September–October (Fig. 7). Anoxia in the central inner basin has been rare since 1976, and hydrogen sulphide was only recorded in 1993 (data not shown). It should, however, be noted that seasonal occurrence of hydrogen sulphide is common in
enclosed, peripheral areas of the inner archipelago (Lännergren and Stehn, 2011). The yearly minimum volume-weighted $O_2$ concentration in the deep water shows a general improving trend (+0.07 mg $L^{-1}$ $yr^{-1}$, $r^2$=0.46, p<0.0001, Fig. 6b). There are also decadal variations, with bad conditions in the 1970´s, better in the 1980´s, worse again in the 1990´s and finally better from 2001. The yearly maximum extent of hypoxic ($O_2$ <2 mg $L^{-1}$) bottom areas was estimated using the hypsographic curve (bottom area per depth layer) of the IA. In the period 1968-1980, hypoxic areas varied greatly, from <10 to 100 % of deep-
water bottoms and 0 to 100 % of 10-20 m bottoms (Fig. 6b). In the 1980´s only minor deep areas were hypoxic (except for 1984), followed in the 1990´s by larger hypoxic areas and then a return to only occasional deep-water hypoxia in the new millennium. From 1994, there has been no hypoxia in the 10-20 m depth interval.

### 3.8 Relationship between $O_2$ and P release in the deep water

The deep-water P release in July-October correlated with the minimum $O_2$ concentration (Fig. 8b). This may partly be due to
the classic mechanism of enhanced P release at low $O_2$ concentrations, but is not necessarily a cause and effect relationship. Both P release and $O_2$ depletion may be a consequence of organic matter degradation, and thus primarily linked to organic matter sedimentation, as is indicated by the correlation between the net P loss in spring and the net P release in summer and autumn (Fig. 8a). To shed light on the cause and effect relationships we also modelled deep-water $O_2$ consumption, which was a function of the seasonal decrease of volume-weighted $O_2$ concentration, the SW $O_2$ inflow (mean model), and the net
$O_2$ input to the deep water from mixing with the 10-20 m layer. The $O_2$ consumption rate (Fig. 7) was low in spring, increased through the summer and peaked in September-October (at 800 tons per month, i.e. 0.8 g $m^{-2}$ $day^{-1}$ or 26 mmol $O_2$ $m^{-2}$ $day^{-1}$ for 32.3 $km^2$ bottom area below 20 m depth).

Theoretically expected P release rates ($P_{O2}$) were calculated from deep-water $O_2$ consumption, assuming mineralization of organic matter of Redfield elemental composition (see methods). Most years, annual $P_{O2}$ calculated from
organic matter decomposition was close to the modelled annual TP and DIP release rates (Fig. 6a-b). Modelled TP and DIP release tended to be higher than $P_{O2}$ in years with low minimum $O_2$ concentrations (e.g. 1984, 1992-93, 1996, 2000, 2008 and 2012) (Fig. 6a and Fig. 9a-b). When the comparison was restricted to July-October, $P_{O2}$ explained less of the P release





(Fig. 6c and Fig. 9 c-d), with largest deviation for September (Fig. 7). Conversely, the P release was generally somewhat lower than $P_{O2}$ in May-June (Fig. 7).

### 3.9 Factors affecting oxygen conditions

The observed $O_2$ minimum correlates with modelled $O_2$ consumption July-October (Fig. 8c, here converted to P release, see above). Since the calculation of $O_2$ consumption depends on the observed $O_2$ minimum concentration, a correlation is expected and should be interpreted with caution. We find a similar correlation between the $O_2$ minimum, which is usually in September, and the $O_2$ consumption in July (data not shown). Although our model results indicate that the $O_2$ consumption rate influences $O_2$ concentrations of the deep water, hypoxia usually developed in years with low $O_2$ inputs to the deep-water, i.e. the sum of $O_2$ inputs from boundary inflow and mixing with intermediate water (Fig. 8c).

Mean $O_2$ consumption in July-October (converted to P mineralization of organic matter, $P_{O2}$) did not correlate with water temperature (Fig. 10a). However, there was a tendency of higher $P_{O2}$ at high temperatures when normalized to the net organic matter sedimentation in spring (as modelled P loss from upper water mass, Fig. 10b), suggesting a higher degradation rate per amount of settled organic matter at higher temperatures. The results were clearer when the same analysis ($P_{O2}$ per P sedimentation as a function of temperature) was done for each month separately, with similar slopes in July, August and September and with decreasing correlation coefficients from July ($r^2$= 0.41, 0.30, 0.18, for each month respectively), and with no correlation in October ($r^2$<0.01) (data not shown). There was no long-term trend in $O_2$ consumption July-October (Fig. 11). The decrease from the 1970's to the 1980's was soon followed by increased $O_2$ consumption in the 1990´s (Fig. 11), possibly influenced by increased temperatures.

### 4 Discussion

Our study highlights the dual function of sediments in a coastal area as sink and source for P and shows that this function strongly depends on temporal scale. Sediments of the IA change from a P sink in spring to a P source in summer and autumn, are an annual net source in some years, especially when there is hypoxia, and has a poor long-term P retention relative to P inputs to the IA. We used two different model approaches to evaluate the internal P fluxes and the P input-export budget. By accounting for the surface salinity gradient from the IA central stations to the boundary, the boundary model gives the most realistic boundary flow, considerably higher than in the mean model, where water turnover will balance mean IA salinity. Despite this difference, both models result in similar input-export P budgets, with differences most years within the uncertainty estimates of the sensitivity tests. We consider the boundary model the most reliable for the input-export budget due to the more realistic boundary flow and the larger data availability at the boundary compared to IA central stations, especially in winter, when net P export is highest. In contrast, water turnover rates of the mean model should be most representative when using the central stations to estimate mean internal fluxes in surface and deep water. Fortunately, the period with highest internal flux rates (April to October) has the best data availability. The similarity of the



input-export budgets of the two models suggest that internal nutrient fluxes in the outer IA section are not fundamentally different from that represented by the central stations.

## 4.1 Long-term phosphorus retention

Two previous studies have estimated long-term P budgets for the Stockholm IA. Using a mass-balance P budget based on
long-term annual means, Karlsson et al. (2010) calculated a P retention of 51 tons per year for the period 1982-95 and 53 tons $yr^{-1}$ for 1996-2007. They also calculated a P retention of 70 tons $yr^{-1}$ based on sediment accumulation rates. Almroth-Rosell et al. (2016) used coupled physical and biogeochemical models to calculate a P retention of 30 tons $yr^{-1}$ for 1990-2012, corresponding to 18 % of the land-derived P loading. In their biogeochemical model, sediment DIP release was $O_2$ sensitive and a fraction of P was assumed to be permanently buried, apparently a nearly constant amount per year. Thus, the
retention estimate was strongly dependent on the chosen model parameters. Compared to other coastal areas a relatively low P retention efficiency (<20 %) is expected for the Stockholm IA, based on its low water residence time (Almroth-Rosell et al., 2016; Nixon et al., 1996; Nielsen et al., 2001).

Our results indicate a near balance between input and export of P, with a tendency to net export during the first two decades following the main P treatment upgrade in the 1970´s. For the last 20-year period, our sensitivity tests suggest a
long-term mean P retention of at most 15 tons per year (Table 3), corresponding to 8 % of summed FW and STP inputs, half the value found by Almroth-Rosell et al. (2016). It is also difficult to reconcile a sediment net burial of 50-70 tons P per year (Karlsson et al. 2010) with our internal flux estimates of net P sedimentation losses from the 0-20 m water layer in March-May of usually less than 50 tons. There is, however, an apparent contradiction between a very low P retention and the presence of accumulation bottoms in the inner archipelago, with a net deposition of sediments (Jonsson et al., 2003). The
difficulty of assessing representative net sediment accumulation rates may contribute to this discrepancy. A major part of the dry matter sedimentation in the archipelago is probably resuspended sediments, with a large fraction having the elemental signature of clay (Blomqvist and Larsson, 1994). Persson and Jonsson (2000) found large variation in yearly sediment accumulation rates, related to the frequency of strong wind events, and concluded that erosion of shallow bottoms contribute much to the sediment accumulation in the deep areas. It may therefore be misleading to assume that sediment accumulation
rates in deep areas represent net loss of P. Moreover, the net P export that seem to have followed the external P load reduction in the 1970´s indicate that sediments were a net source of P, possibly as a legacy of the previous high P loading. Degradation of organic P, combined with limited retention of Fe-P in surface sediments during poor $O_2$ conditions, may have promoted this excessive P release. However, relative to the previously very high P loading, with accumulated inputs 1968-75 as high as 3900 tons from STPs only (see Table 1), legacy P release is small. Conditions favouring large P export also at
high P loading seem to have limited the build-up of a large sediment legacy P stores. (Fig. 5c). Sediment cores from the 1990´s indicate no remaining legacy of carbon in IA sediments (Jonsson et al., 2003).

The now stabilized, near neutral P balance means that there still is little P retention, in spite of the improved $O_2$ conditions. Although oxidation of surface sediments can increase short-term P retention due to accumulation of Fe-





oxyhydroxides, many studies indicate little effect of this P pool on long-term P retention, which depends on the amount of P permanently buried in the deeper anoxic parts of the sediments (Jensen et al., 1995; Lukkari et al., 2009; Rydin et al., 2011). Recent results indicate that a persistent Fe(II)-P mineral can form in sediments of the Bothnian Sea (Slomp et al., 2013) and to some extent also in the Baltic proper (Reed et al., 2016). To which extent formation of Fe(II)-P minerals have contributed

to long-term retention in the Stockholm IA is unknown, but the low overall P retention suggest this process is currently not very significant. Possibly, preferential Fe(II) sulphide formation in IA sediments competes with Fe(II)-P formation at the current organic loading (Reed et al. 2016).

Norkko et al. (2011) have suggested that the recent (post-2005) increase of the invasive polychaetes *Marenzelleria* spp. in the IA may have enhanced the oxygenation of the sediment and hence increased the binding of P in the sediments,

presumably as Fe-oxyhydroxides. We found neither lower P release rates nor increased P retention in these years. However, our results are not directly comparable to those of Norkko et al. (2011), who used data from the inner part of the IA only.

## 4.2 Seasonal phosphorus budget in surface water

In spring, there is net internal TP loss (largely as DIP) from the whole water mass of the IA, due to larger losses from the upper layers (0-20m) than net P inputs to the deep water >20m (Fig. 7). The DIP and TP loss was associated with the

phytoplankton spring bloom that is typical for temperate coastal areas (Blomqvist and Larsson, 1994; Testa and Kemp, 2008; Staehr et al., 2017). In the IA, diatoms and dinoflagellates, known to have high sedimentation rates, dominate the spring bloom (Lännergren and Stehn, 2011). The incorporation in higher trophic levels probably contributes only to a small extent to the P loss, since zooplankton are included in the TP analysis and the build-up of fish biomass mainly occurs in summer (Hjerne and Hansson, 2002). P uptake by benthic vegetation likely contributes little to the overall P budget, since the deep

IA has small bottom areas suitable for macrophytes, and large algae, such as bladderwrack (*Fucus vesiculosus*), are absent from this low-salinity area.

In summer and autumn, net P losses from the 0-20 m surface layer decrease while net P inputs, mainly of DIP, increase in the deep water. This requires a net internal P source, which we interpret as net sediment P release, and increases the P concentration in all water layers of the IA (Fig. 7). However, there is no net export of P from the IA until the winter

months.

Our model approach can only give net rates for each compartment. That the annual net P loss from the upper 0-20 meters is relatively small in some years (Fig. 6a) probably does not reflect lower gross sedimentation rates, but substantial regeneration of P also from shallow sediments. Such P release in summer can explain the much larger variability of 0-20 m layer net P loss for the full year than for the spring only, when P recycling from sediments should generally be low. Bottoms

0-20 m of which only a small proportion are accumulation bottoms (Jonsson, 2003), account for 69 % of the area in the inner archipelago, and are probably particularly important for regeneration of P in years when transport to deeper accumulation bottoms is less effective, or when hypoxia affect also shallow areas, e.g. in 1992 and 1993. Sedimented matter is probably efficiently transported to deep accumulation bottoms, as most of the net P sedimentation in spring seems to be regenerated in



the deep water in summer and autumn (Fig. 6). However, it remains difficult to estimate the contribution of gross sedimentation P losses from the surface layer to the deep water in summer.

## 4.3 Seasonal phosphorus and oxygen budgets in deep water

Our monthly P-budgets for the deep-water indicate a more or less continuous internal DIP source, with a clear increase in

summer and a peak in September, similar to what Jensen et al. (1995) found in Aarhus Bay. The budget for the whole water column shows a net loss of P to the sediments in spring, suggesting deep-water sediments are a net sink for P in spring and early summer. The delayed net P input to the whole water column in summer and autumn suggests that most of this internal P source in the deep water is regeneration from seasonal storage of P in deep sediments, but a P redistribution by transport of shallow sediments to the deep water can also contribute. On an annual basis, there was a strong correlation between deep

water $O_2$ consumption rates and TP and DIP release, close to Redfield proportions (Fig. 9a), indicating that P release is mainly short-term (within-year) recycling of sedimented organic P. Moreover, net P release in the deep water correlated with net P spring sedimentation (Fig. 8a, Fig. 7), suggesting that a large share of the net P release in the deep water in summer and autumn is a seasonal recycling of P from the spring bloom.

The summer increase in P flux from sediments to the deep-water starts in June (Fig. 7), despite $O_2$ concentrations in

June and July well above hypoxic levels, even in the deepest bottom water. In August and September, P release continues to increase also in years with relatively good $O_2$ conditions. These observations, and the close coupling between deep-water P release and respiration rates, indicate limited classical Fe-P trapping (Einsele, 1936; Mortimer, 1941) in the oxic surface sediments. Similarly, field studies in the Finnish archipelago have shown high DIP release also under oxic conditions, with limited Fe-P formation as the suggested explanation (Lehtoranta and Heiskanen, 2003; Lehtoranta et al. 2009). In relatively

sulphate-rich coastal sediments, Fe can be immobilized as solid, unreactive FeS or $FeS_2$ in anoxic layers of the sediment, where sulphate-reducing bacteria produce hydrogen sulphide (Blomqvist et al., 2004). As a result of low Fe availability relative to the production of dissolved P from organic matter decomposition, the rate of P release to the water mass is then mainly determined by the rate of remineralisation (Caraco et al., 1990; Hupfer and Lewandowski, 2008). We found that P release was somewhat lower than expected from $O_2$ consumption in spring and higher in summer. This suggests some early

season trapping of P in Fe-P complexes, which partly dissolve in late summer, when the redox-cline in the sediment becomes shallower and higher temperatures increase reduction rates of Fe-oxyhydroxides (Jensen et al., 1995; Jilbert et al., 2011). Jensen et al. (1995) found that P sedimented in spring was largely retained as Fe-bound P until September-October, when half the annual P release occurred. For the Arkona basin of the Baltic Sea, Reed et al. (2011) estimated that sediment DIP release at oxic conditions approximately doubled from winter to summer (0.05 - 0.1 mmol m$^{-2}$ day$^{-1}$), recycling much of the

P mainly deposited as organic matter in spring. Aerobic degradation of organic matter was the dominant P mineralization process, but P release was largely mediated via rapid Fe-oxyhydroxide turnover. Elevated temperatures in the IA after 1990 may have increased anaerobic respiration and sulphide production (Bågander, 1977), promoting P release by decreasing the available Fe due to Fe-sulphide formation.





Lehtoranta et al. (2009) suggested that the Fe-dependent P-binding capacity of marine sediments can be negatively affected by previous anoxic events, if much of the Fe pool has become permanently bound as sulphides and is unavailable for P-binding upon reoxidation. Effective P-binding capacity will then be regained only slowly, upon replenishment of the Fe-pool. Early investigations show that IA sediments were anoxic long before the rapidly increasing loads of organic matter and nutrients in the mid-20th century (Brattberg, 1986). Hydrogen sulphide was recorded as early as 1909 and was present in the inner parts of the IA (Norrström – Höggarnsfjärden) on 10 out of 10 sampling occasions in 1909–1913 and in 40 out of 50 occasions in 1930–1944 (Anon., 1945). Dredging of the strait Oxdjupet, from 8 to 10.5 m in 1919 and to 12 m in 1929 apparently had little effect, but dredging in 1950 to the present 18 m improved $O_2$ conditions. Sediment cores show increasing lamination frequency until 1950, followed by decreased frequency, indicative of improved $O_2$-conditions and more benthic fauna (Jonsson et al., 2003). In 1970–1980, lamination frequency increased again, remaining high at least until the mid-1990´s (Jonsson et al., 2003). In 2008, surface sediments were oxidized (Karlsson et al. 2010). How the history of anoxic events has affected later Fe- and P-cycling in the Stockholm IA is not well understood, but the short water turnover time of the IA suggests that continuous inputs of Fe should be relatively more important for formation of sediment Fe(III)-P complexes than the internal pool. Monitoring data from the Lake Mälaren outlet Norrström show that the Fe transport is substantial, ~400 tons per year, corresponding to ~70 µg $L^{-1}$ (measurements by Sveriges Lantbruksuniversitet, www.slu.se/institutioner/vatten-miljo/datavardskap/).

### 4.4 Phosphorus release at low oxygen concentrations

In our model, all inflowing SW is well mixed in the deep water and the estimates of deep-water P release are a direct function of the deep-water turnover rates. Thus, if the deep-water becomes partly stagnant and the SW inflow partly mixes into shallower (10-20 m) water layers, the model will somewhat overestimate deep-water P release rates. Since stagnation may contribute to hypoxia, this risk is higher in years with hypoxia and therefore the differences in P release between hypoxic and oxic years are more likely overestimated than underestimated. However, variable deep-water turnover rates will have similar effect on deep-water P release and $O_2$ consumption. Thus, the finding of high P release rates relative to theoretical P remineralization (based on $O_2$ consumption) in years with low $O_2$ concentrations (Fig. 9) is not affected by this. Likewise, the P retention estimate for the entire IA is independent of modelled deep-water P cycling.

The particularly high P release in years with deep water hypoxia (Fig. 9) suggests that classical redox-dependent release of Fe(III)-bound P contributed significantly to P release in those years. We also found particularly high P net export in years when hypoxia affected intermediate water depths (e.g. in 10-20 m layer in 1992-1993), suggesting unusually large P release from sediment Fe-P stores in these water layers. This is in contrast to the net retention in more $O_2$-rich years, when Fe-P pools probably were restored. Puttonen et al. (2014) estimated the reserve of potentially mobile P in Baltic Sea archipelago and Stockholm inner archipelago surface sediments to 3.5 g $m^{-2}$, stored predominantly in redox-sensitive Fe-bound forms on accumulation bottoms. This P reserve is similar to our estimated yearly net P release from deep IA bottoms (up to 3.1 g $m^{-2}$ $yr^{-1}$ and a mean of 2.1 g $m^{-2}$ $yr^{-1}$) and to the P release of 2.7 g $m^{-2}$ $yr^{-1}$ estimated by Rydin et al. (2011) in




Torsbyfjärden in the outer IA. Our estimate of the P amount released in the deep water (67 tons $yr^{-1}$) is not sensitive to the defined depth of the deep-water. However, with 15-57 m instead of 20-57 m as deep-water layer the sediment area increases by 40 % and the area-specific mean P release is 1.5 g $m^{-2}$ $yr^{-1}$. Still, Fe-P stores (Puttonen et al. 2014) are not very large relative to yearly P recycling from deep-water sediments of the IA and the notion that such mobile P stores must have accumulated over decades of high anthropogenic P loads is questionable for the Stockholm IA.

### 4.5 Factors affecting oxygen conditions

Our model indicates that although $O_2$ consumption is an important factor for deep-water $O_2$ concentrations in the Stockholm IA, hypoxia is particularly prone to develop in years with low $O_2$ inputs to the deep water, i.e. inputs with boundary inflow and mixing of deep water with intermediate water (Fig. 8c). This was evident already in the 1970's. Despite high organic matter loads to the deep water (judging from the P budget), our model results indicate that high $O_2$ inflow prevented widespread hypoxia in 1969 and 1970.

We find no long-term trend (1968-2015) in July-October $O_2$ inflow to the IA deep water from the middle archipelago (data not shown). However, there is a trend of increasing $O_2$ input to the deep water from mixing with intermediate water layers (10-20m), where $O_2$ concentrations have increased. STP discharges of $O_2$ consuming substances, mostly ammonium, was ~15000 tons $O_2$-equivalents $year^{-1}$ before sewage treatment was improved in the mid-1990´s and decreased to ~3000 tons by 2000 (Lännergren and Stehn, 2011). We made no $O_2$ budget for the intermediate water, but considering mixing rates, the improvement was large enough to contribute to the disappearance of hypoxia in the 10-20 m depth layer. In addition, the weakening of the stratification that occurred around 1990 promoted vertical mixing and thus $O_2$ transport to the intermediate layer.

Organic matter inputs ultimately control the total $O_2$ consumption in the deep water. However, temperature-dependent aerobic and anaerobic microbial processes should influence $O_2$ consumption rates. We found increasing $O_2$ consumption rates from May to September or October, when the temperature increased from ~2 to 9 °C. The maximum $O_2$ consumption, in September (26 mmol $O_2$ $m^{-2}$ $d^{-1}$) corresponds well with summer $O_2$ consumption rates measured in Danish estuaries (Ærtebjerg et al., 2003) and Finnish archipelago areas (Kauppi et al., 2017). Ærtebjerg et al. (2003) found that temperature explained most of the within-year variation in $O_2$ consumption. In the Stockholm IA, decomposition rates of organic material should have increased due to higher temperature from 1990 on (Fig. 3c). We find that $O_2$ consumption rate in July-October normalised to organic matter input in spring increased as postulated (Fig. 10a). As expected, the correlation based on monthly data was strongest in early summer, when most organic matter from the spring bloom is undegraded. We find no long-term trend in total $O_2$ consumption in July-October. Lowered organic matter deposition probably explains the decreased $O_2$ consumption in the late 1970's. However, in the 1990´s increased $O_2$ consumption due to higher temperatures probably counteracted this improvement and contributed to the renewed spread of hypoxia.





## 5. Conclusions

We conclude that the seasonal increase of water column P concentrations in the Stockholm inner archipelago in summer and autumn is due mainly to internal P loading from the sediments. Most of this P release is simply a recycling of P deposited by the sedimentation of the spring bloom. Legacy P release from historical P deposits, presumably organic P, was probably

restricted to the first 10-20 years after the major P load reduction, when model results suggest sediments were net P sources. The long-term P balance has now stabilized, with means of our sensitivity analyses close to neutral balance (i.e. export equals inputs) in the last two ten-year-periods, indicating that internal loading from legacy P has ceased. The sensitivity analysis suggest a retention <15 tons per year, or 8 % of land-based inputs. The external P inputs are either exported without seasonal delay, or are efficiently recycled from sediments in summer and autumn despite the improved $O_2$ conditions, and

then exported in winter. The stabilized P balance means that further water quality improvements cannot be expected for the Stockholm IA unless nutrient load is further decreased, or that long-term sediment P retention is improved.

The tightly coupled P release and $O_2$ consumption processes show that the often-observed correlation between water $O_2$ concentrations and P (e.g. Norkko et al., 2011) is only partly a cause-and effect relationship, since both $O_2$ and P are linked to organic matter degradation (cf. Hupfer and Lewandowski, 2008; Reed et al., 2011). Thus, Fe compounds

apparently cannot prevent P release into the overlying oxic water. In winter and spring, when $O_2$ concentrations peak, some temporary storage of Fe-P probably occurs, which is released when Fe is increasingly bound to sulphides in summer, as the oxic upper sediment layer gets thinner because of increasing $O_2$ consumption rates at the higher temperatures. Consistent with the classical view of redox-dependent P release, summer and autumn hypoxia triggered elevated P release, often resulting in a low annual P retention of the area, when sediments at intermediate depth were affected by hypoxia.

We found that hypoxia in the IA deep water was influenced by the size of the spring bloom but also by temperature, deep-water renewal rates, and the $O_2$ concentrations in the water ventilating the deep-water. The generally improving $O_2$ conditions of the IA are probably linked to lowered sedimentation of organic matter (especially in the 1970s), to lowered STP loading of ammonium (from the late 1990´s) and enhanced vertical mixing (from the 1990´s). The enhanced mixing has also contributed to higher deep-water temperature and respiration rates. More detailed studies are needed to clearly separate

the complex factors influencing the $O_2$ conditions in the Stockholm IA.

The impact of hypoxia-controlled internal P loading on long-term ecosystem productivity depends on the size of this P pool relative to external P loading and the P inventory in the water mass. In the Baltic Sea as a whole, hypoxia clearly affects the P-inventory of the water mass over long periods (Conley et al., 2002). The Baltic Sea P inventory of ~400,000 tons fluctuates by 100,000 tons between years (Jilbert et al., 2011), considerably more than the yearly P loading from land of

~28,000 tons (HELCOM, 2011). In the well-flushed Stockholm IA, the annual mean P inventory in the water mass is currently ~46 tons, fluctuating by only a few tons between years and 30 tons seasonally, much less than the yearly P loading from land of 190 tons. Release of P during hypoxic events will largely be exported in winter and will have very limited long-term effects locally.

Measures aimed at artificially increasing P trapping in sediments, for example by aluminium treatment, can be successful in archipelago bays with restricted water exchange (Rydin et al., 2017). The short P turnover time in the Stockholm IA, suggests that here sediment treatment must enhance long-term annual P retention efficiency, probably by recurrent treatment, to have long-term effects on local water quality. Even so, treatment of IA sediments primarily would not

bind legacy P, but rather accumulate P from the Baltic Sea and Lake Mälaren and to minor extent the relatively small current P inputs from STPs, suggesting control of external P inputs is preferable. However, not only management of P should be considered, since further intensifying P limitation in the IA would lead to larger N export to N-limited parts of the archipelago (cf. Brattberg, 1986; Paerl, 2009; Lännergren and Stehn, 2011).

## Data availability

Monitoring data are deposited at the Swedish Meteorological and Hydrological Institute (www.smhi.se). SMHI also provides runoff data.

## Acknowledgements

Data were kindly provided by Stockholm Vatten AB (now Stockholm Vatten och Avfall). Model development was initiated in the EU-project SPICOSA, and was also financially supported by Sydvästra Stockholmsregionens va-verksaktiebolag

(SYVAB). Data compilation and model analysis was supported by grants from Svealands kustvattenvårdsförbund (SKVVF). Data digitizing was financed by the Swedish EPA, Stockholm Vatten AB, and SKVVF. We thank Ragnar Elmgren for valuable comments on the manuscript.

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

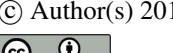


**Table 1.** Period means of yearly water discharge (Q), total phosphorus (TP) load and volume weighted input TP concentrations (±standard deviation). The volume weighted TP concentration in the whole water mass and in the surface layers of the model area are also shown. LM=Lake Mälaren, LR=local runoff, P=precipitation, STP=sewage treatment plants, IA=inner archipelago.

| | | 1968–1975 | 1976–1985 | 1986–1995 | 1996-2005 | 2006-2015 |
|---|---|---|---|---|---|---|
| $Q_{LM}$ | $Mm^3 yr^{-1}$ | 4027 ±884 | 5414 ±1717 | 5004 ±939 | 4711 ±1682 | 5543 ±1189 |
| $Q_{LR}$ | $Mm^3 yr^{-1}$ | 83 ±0 | 88 ±16 | 85 ±21 | 76 ±20 | 88 ±17 |
| $Q_P$ | $Mm^3 yr^{-1}$ | 50 ±10 | 58 ±8 | 58 ±8 | 54 ±7 | 59 ±10 |
| $Q_{STP}$ | $Mm^3 yr^{-1}$ | 158 ±7 | 162 ±10 | 189 ±18 | 182 ±14 | 195 ±14 |
| $TP_{LM}$ | $tons\ yr^{-1}$ | 228 ±108 | 223 ±69 | 143 ±43 | 138 ±59 | 143 ±27 |
| $TP_{LR}$** | $tons\ yr^{-1}$ | 8 ±0 | 9 ±2 | 8 ±2 | 8 ±2 | 9 ±2 |
| $TP_{STP}$ | $tons\ yr^{-1}$ | 485 ±272 | 74 ±12 | 61 ±12 | 28 ±3 | 33 ±6 |
| $TP_{LM}$ | $\mu g\ L^{-1}$ | 55 ±18 | 42 ±6 | 28 ±4 | 29 ±4 | 26 ±1 |
| $TP_{LM+LR+STP}$ | $\mu g\ L^{-1}$ | 167 ±69 | 57 ±15 | 40 ±5 | 35 ±3 | 32 ±2 |
| $TP_{IA}$ 0-57 m | $\mu g\ L^{-1}$ | 118 ±28* | 49 ±4 | 39 ±3 | 32 ±2 | 30 ±2* |
| $TP_{IA}$ 0-10 m | $\mu g\ L^{-1}$ | 123 ±38* | 45 ±4 | 36 ±3 | 30 ±3 | 28 ±2* |
| $TP_{IA}$ 0-20 m | $\mu g\ L^{-1}$ | 122 ±32* | 48 ±3 | 38 ±3 | 31 ±2 | 29 ±2* |
| $TP_{IA}$ 20-57 m | $\mu g\ L^{-1}$ | 93 ±11* | 55 ±7 | 43 ±5 | 40 ±5 | 39 ±3* |

5    * Years 1975 or 2011 excluded.

**A mean local runoff TP concentration of 100 µg/l is assumed, see methods.



**Table 2.** Period means of modeled yearly water inflow (Q), total phosphorus (TP) input and volume weighted input TP concentrations (±standard deviation). SW=inflowing salt water from middle archipelago, MM=mean model, BM= boundary model.

|  |  | 1968–1975 | 1976–1985 | 1986–1995 | 1996-2005 | 2006-2015 |
|---|---|---|---|---|---|---|
| **Q$_{SW-MM}$** | Mm$^3$ yr$^{-1}$ | 3126 ±443* | 3189 ±611 | 3684 ±643 | 3973 ±554 | 4109 ±419 |
| **Q$_{SW-BM}$** | Mm$^3$ yr$^{-1}$ | 5608 ±828* | 5441 ±935 | 7095 ±960 | 7717 ±890 | 6963 ±767 |
| **TP$_{SW-MM}$** | tons yr$^{-1}$ | 110 ±20* | 90 ±15 | 93 ±15 | 100 ±12 | 99 ±18* |
| **TP$_{SW-BM}$** | tons yr$^{-1}$ | 199 ±41* | 159 ±32 | 186 ±33 | 201 ±28 | 175 ±26* |
| **TP$_{SW-MM}$** | µg L$^{-1}$ | 35 ±4* | 28 ±2 | 25 ±2 | 25 ±3 | 24 ±2* |
| **TP$_{SW-BM}$** | µg L$^{-1}$ | 35 ±4* | 29 ±2 | 26 ±2 | 26 ±3 | 25 ±2* |

* Years 1975 or 2011 excluded.





**Table 3.** Effects of sensitivity tests of net P export (tons per year) from the inner archipelago (IA) with boundary model (0-6), and mean model (7). Positive values indicate that the IA accumulates (is net a sink for) P, negative values that it is a net source of P. Year 2011 is excluded. Sensitivity tests included 1. Variable freshwater flow with Lake Mälaren ($Q_{LM}$), 2. Recruitment depth of inflowing salt water (SW) from the middle archipelago (affecting salinity and P), 3. Sewage treatment plant (STP) discharge of P, 4. Water P concentrations in the IA and Lake Mälaren (LM), 5. Deeper layer for water export from the IA, 6. Non-adjusted data from central IA stations (see methods), 7. Mean model.

| Sensitivity test | 1976–1985 | 1986–1995 | 1996-2005 | 2006-2015 | 1976–2015 |
|---|---|---|---|---|---|
| 0. Default boundary model | -25.8 | -19.6 | 0.1 | 3.8 | -10.7 |
| 1a. $Q_{LM}$ -5 % | -20.7 | -15.7 | 1.8 | 5.3 | -7.7 |
| 1b. $Q_{LM}$ +5 % | -30.9 | -23.4 | -1.5 | 2.2 | -13.8 |
| 2a. SW 12–20 m | -51.0 | -53.6 | -32.9 | -23.8 | -40.7 |
| 2b. SW 16–24 m | -39.1 | -39.9 | -19.3 | -13.8 | -28.4 |
| 2c. SW 24–32 m | -18.3 | -7.2 | 12.3 | 15.1 | 0.1 |
| 3a. STP -10 % | -33.2 | -25.7 | -2.7 | 0.5 | -15.7 |
| 3b. STP +10 % | -18.4 | -13.4 | 2.9 | 7.0 | -5.8 |
| 4a. TP conc. IA, LM -10 % | -8.0 | -1.3 | 13.8 | 15.1 | 4.7 |
| 4b. TP conc. IA, LM +10 % | -43.6 | -37.8 | -13.6 | -7.6 | -26.1 |
| 5. Export depth 0-10 m | -48.1 | -28.0 | -5.4 | 4.9 | -19.8 |
| 6. Non-adj. IA Salinity/ TP | -23.5 | -19.0 | 1.0 | 4.0 | -9.7 |
| 7a. Mean model with boundary P conc. in export calculation. | 7.9 | 3.5 | 10.3 | 10.1 | 7.9 |
| 7b. Mean model | 1.2 | -16.2 | -8.4 | -7.9 | -7.8 |





**Table 4.** Effects of sensitivity tests on deep layer (>20 m) P release (tons per year) in the inner archipelago (IA) using the mean model. Year 2011 is excluded. Sensitivity tests included 1. Variations in freshwater flow with Lake Mälaren ($Q_{LM}$), 2. Recruitment depth of inflowing salt water (SW) from the middle archipelago (affecting salinity and P), 3. Sewage treatment plant (STP) discharge of P, 4. Water P concentrations in the IA and Lake Mälaren (LM), 5. Deeper layer for water export from the IA, 6. Non-adjusted data from central IA stations (see methods), 7. Use of boundary P concentration for export from the IA.

|  | 1976–1985 | 1986–1995 | 1996-2005 | 2006-2015 | 1976-2015 |
|---|---|---|---|---|---|
| **0. Default mean model** | 70.4 | 59.5 | 62.2 | 71.6 | 66.2 |
| **1a. $Q_{LM}$ -5 %** | 67.4 | 57.3 | 59.7 | 68.9 | 63.6 |
| **1b. $Q_{LM}$ +5 %** | 73.4 | 61.7 | 64.7 | 74.2 | 68.8 |
| **2a. SW 12–20 m** | 83.9 | 75.7 | 72.9 | 82.8 | 79.0 |
| **2b. SW 16–24 m** | 78.0 | 69.1 | 69.4 | 79.6 | 74.3 |
| **2c. SW 24–32 m** | 66.0 | 53.5 | 57.3 | 66.0 | 61.0 |
| **3a. STP -10 %** | 70.4 | 59.6 | 62.2 | 71.6 | 66.3 |
| **3b. STP +10 %** | 70.3 | 59.4 | 62.2 | 71.5 | 66.2 |
| **4a. TP conc. IA, LM -10 %** | 62.9 | 53.5 | 55.9 | 64.3 | 59.4 |
| **4b. TP conc. IA, LM +10 %** | 77.8 | 65.5 | 68.5 | 78.8 | 73.0 |
| **4c. TP conc. IA -10 %** | 54.7 | 45.1 | 46.9 | 55.4 | 50.7 |
| **4d. TP conc. IA +10 %** | 86.1 | 73.9 | 77.6 | 87.7 | 81.8 |
| **5. Export depth 0-10 m** | 89.2 | 71.6 | 74.3 | 85.0 | 80.3 |
| **6. Non-adj. IA Salinity/ TP** | 62.6 | 54.0 | 55.0 | 65.7 | 59.5 |
| **7. Boundary P conc. in export** | 70.4 | 59.5 | 62.2 | 71.5 | 66.2 |




**Figure legends**

**Figure 1.** Maps showing the location of the inner Stockholm archipelago (IA), in the NW Baltic Proper. Lake Mälaren with drainage area is shown in the top left map and its outlet Norrström in Stockholm (LM) in the lower map. The small local drainage area of the IA is shown in the lower map. The central stations (A, AV, H, L, K) were used to represent the IA. The

major water exchange between the IA and middle archipelago is through Oxdjupet (O). Data from Trälhavet (T) were used to represent inflowing water. Surface water data from Solöfjärden (S), Oxdjupet (O) and Trälhavet Fiskare (TF, weekly station for surface water) were used to represent outflowing water of the boundary model. Depths (m) in maps are spline interpolated from nautical chart data (scale 1:25000, published 1984).

**Figure 2.** Schematic illustration of the box model for water flows ($Q$), salt amounts ($S$) and salinities ($Sal$). Salt is transported from the middle archipelago through Oxdjupet strait ($Q_{in}$ $Sal_{in}$ according to Eq. 2) and forms the inner archipelago deep water, with salt content $S_D$. Upwelling water (equivalent to $Q_{in}$) transports salt to the above layers and mixing ($Q_{D\_mix}$, $Q_{M\_mix}$ and $Q_{S\_mix}$) relocates salt between the layers.

**Figure 3.** Time series 1968-2015 of **(a)** Volume weighted mean salinity of the whole water mass and bottom water (20–57 m) of the inner archipelago (IA) and salinity of the assumed inflowing water (SW) from the middle archipelago at 20-30 m depth just outside the boundary strait (Trälhavet, T in Fig.1). **(b)** Salinity stratification in the inner archipelago. The black contours show integer values 2-6. The colour scale has a resolution of 0.25. **(c)** Seasonal (July-October) mean temperature at 0-8 m, 12-20 m and 24-32 m depth at station Koviksudde (K). **(d)** Volume weighted concentrations of total phosphorus (TP)

and dissolved inorganic phosphorus (DIP) 0-20 m. **(e)** Volume weighted concentrations of TP and DIP in the deep layer, >20 m.

**Figure 4. (a)** Annual mean fresh water input (including STPs) and **(b)** modelled inflow from Middle archipelago (Baltic Sea) to the inner archipelago. Lake Mälaren contributes 93.6 ±2.1 % (mean and SD for all years) and STPs 3.6 ±1.2 % of the

total freshwater inputs.

**Figure 5.** Annual water flows, total phosphorus (TP) inputs, TP export and net TP input-export. **(a)** Annual TP load from sewage treatment plants (STP), from fresh water runoff from Lake Mälaren plus the local catchment (FW), and **(b)** from modelled inflow of middle archipelago deep water (Trälhavet). **(c)** Total annual external TP input and export, and net input-

export, using the boundary model. Positive values indicate that the inner archipelago accumulates P and negative values that it is a net source of P. **(d)** Net input-export according to boundary model (enlargement of c) with error bars showing maximum and minimums from sensitivity test. Dots show net input-export using the mean model.





**Figure 6.** Internal total phosphorus (TP) fluxes and oxygen ($O_2$) conditions. **(a)** Annual net TP release in deep layer (>20 m), annual net TP loss from 0-20 m, net TP balance for whole water mass, and calculated P remineralization from organic matter (from modelled $O_2$ consumption). Positive net TP balance indicates net TP release from sediments to water mass and negative balance indicates net TP loss from water to sediments. **(b)** As in (a) but net TP release in deep layer (>20 m) for July-October and net TP loss from 0-20 m in March-May **(c)** Minimum volume-weighted $O_2$ concentration in deep water and maximum percent hypoxic ($O_2$ <2 mg $L^{-1}$) bottom area in deep water and 10-20 m layer (July-October).

**Figure 7.** Monthly means and changes 1996-2015 (2011 excluded) of **(a)** total phosphorus (TP) stock in whole water mass and deep water and the monthly change **(b)** TP input and export with boundary model, **(c)** net TP release in deep water (>20m) and net TP losses in upper water layers 0-20m and calculated P remineralization (from modeled $O_2$ consumption), with mean model, **(d)** deep-water temperature, $O_2$ consumption and minimum $O_2$ concentration. Error bars show standard deviation.

**Figure 8. (a)** Yearly net total phosphorus (TP) release (tons) in deep water (>20m) in July-October, as a function of net TP loss (tons) from the layers 0-20 m in Mar-May 1969-2015 (1970 excluded as outlier [P loss 222 tons], 1975 excluded due to lack of data and 2011 due to unreliable data). Colour-code shows mean bottom water temperature Jul-Oct (station K, 24-32m, see Fig. 3). There is a significant linear regression, y=19.8 + 0.55x, $r^2$=0.38, p<0.0001. Only the 1:1 line is shown in the graph. **(b)** Yearly net TP release in deep water July-October (tons) as a function of the minimum oxygen ($O_2$) concentration (mg $L^{-1}$). Colour-coded as in (a). There is a significant linear regression, y=73.2 - 8.50x, $r^2$=0.48, p<0.0001. **(c)** Minimum $O_2$ concentration (mg $L^{-1}$) as a function of $O_2$ consumption rate, converted to expected P release (see methods). Colour-code shows model-calculated $O_2$ input to the deep water normalized to the mean $O_2$ input for all years. A value >1 indicates a higher $O_2$ input than the mean value for all years, and value <1 indicates a lower $O_2$ input than the mean for all years. The linear regression is y=6.31-0.10x, $r^2$=0.23.

**Figure 9.** Net total phosphorus (TP, **a, c**) and dissolved inorganic phosphorus (DIP, **b, d**) release in tons per year in the deep water layer (>20 m) versus expected P remineralization (calculated from oxygen consumption) for full years **(a, b)** or July-October **(c, d)** 1969-2015 (1975 and 2011 excluded). The 1:1 line is shown for comparison. The data set is colour-coded according to minimum oxygen ($O_2$) concentration Jul-Oct. Linear regressions were all significant (not shown in graphs), **a:** y=8.18+1.19x, $r^2$=0.42, p<0.0001 for all values, y=11.1+0.91x, $r^2$=0.73, p<0.0001 for $O_2$ minimums >4; **b:** y=4.79+1.17x, $r^2$=0.30, p<0.0001 for all values, y=7.71+0.88x, $r^2$=0.64, p<0.0001 for $O_2$ minimums >4; **c:** y=-2.69+1.70x, $r^2$=0.46, p<0.0001 for all values, $r^2$=10.6+0.91x, $r^2$=0.20, p<0.05 for $O_2$ minimums >4; **d:** y=-8.55+1.89x, $r^2$=0.48, p<0.0001 for all values, y=6.27+1.04x, $r^2$=0.20, p<0.05 for $O_2$ minimums >4).





**Figure 10. (a)** Yearly oxygen consumption July-October 1969-2015 (1975 and 2011 excluded) recalculated to theoretical deep-water P remineralization, as a function of temperature. The data set is colour-coded according to minimum oxygen concentration Jul-Oct. (y=28.3-0.31x, $r^2$=0.006, p=0.60)

**(b)** as in (a) but theoretical P remineralization is normalized to P loss from 0-20 m layer in March-May. (y=-0.030+0.13x, $r^2$=0.25, p<0.001).

**Figure 11.** Oxygen consumption July-October in the Stockholm inner archipelago deep-water 1968-2015 calculated with the mean model.





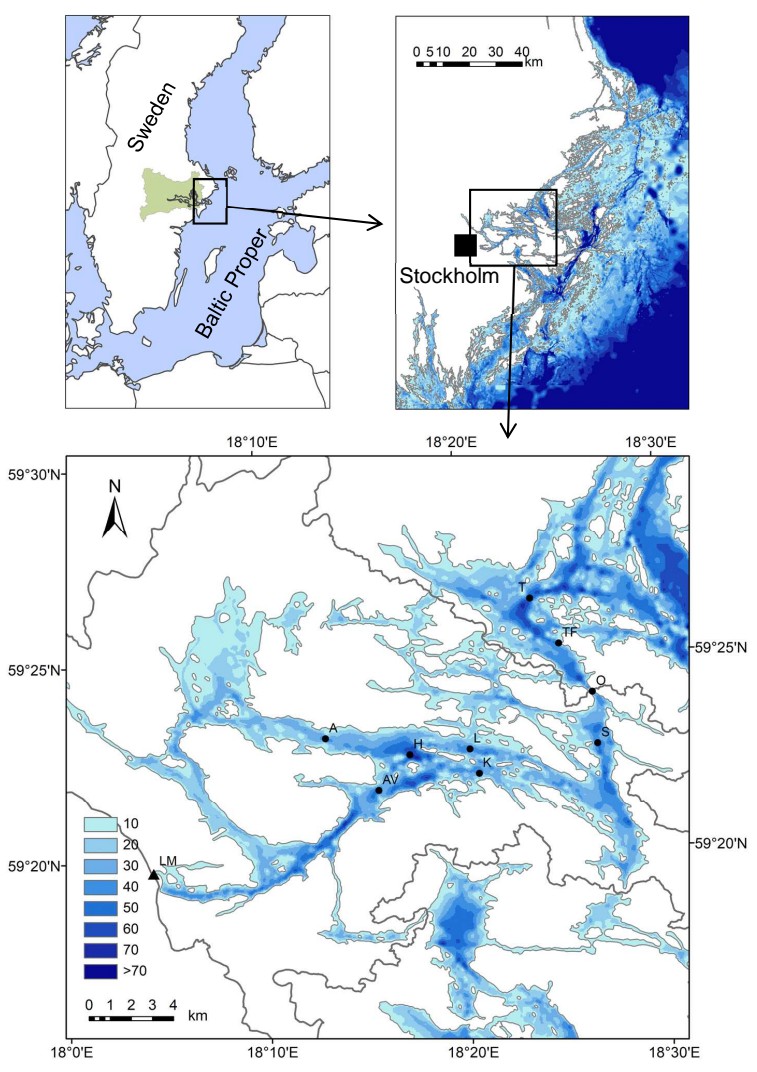

**Figure 1.**



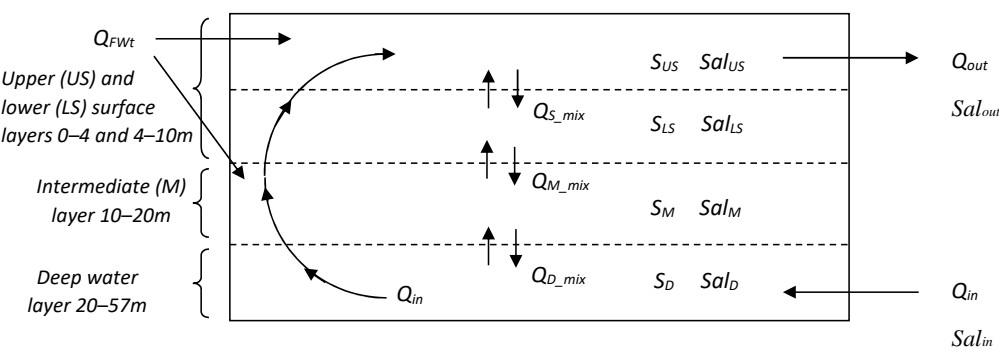

**Figure 2.**





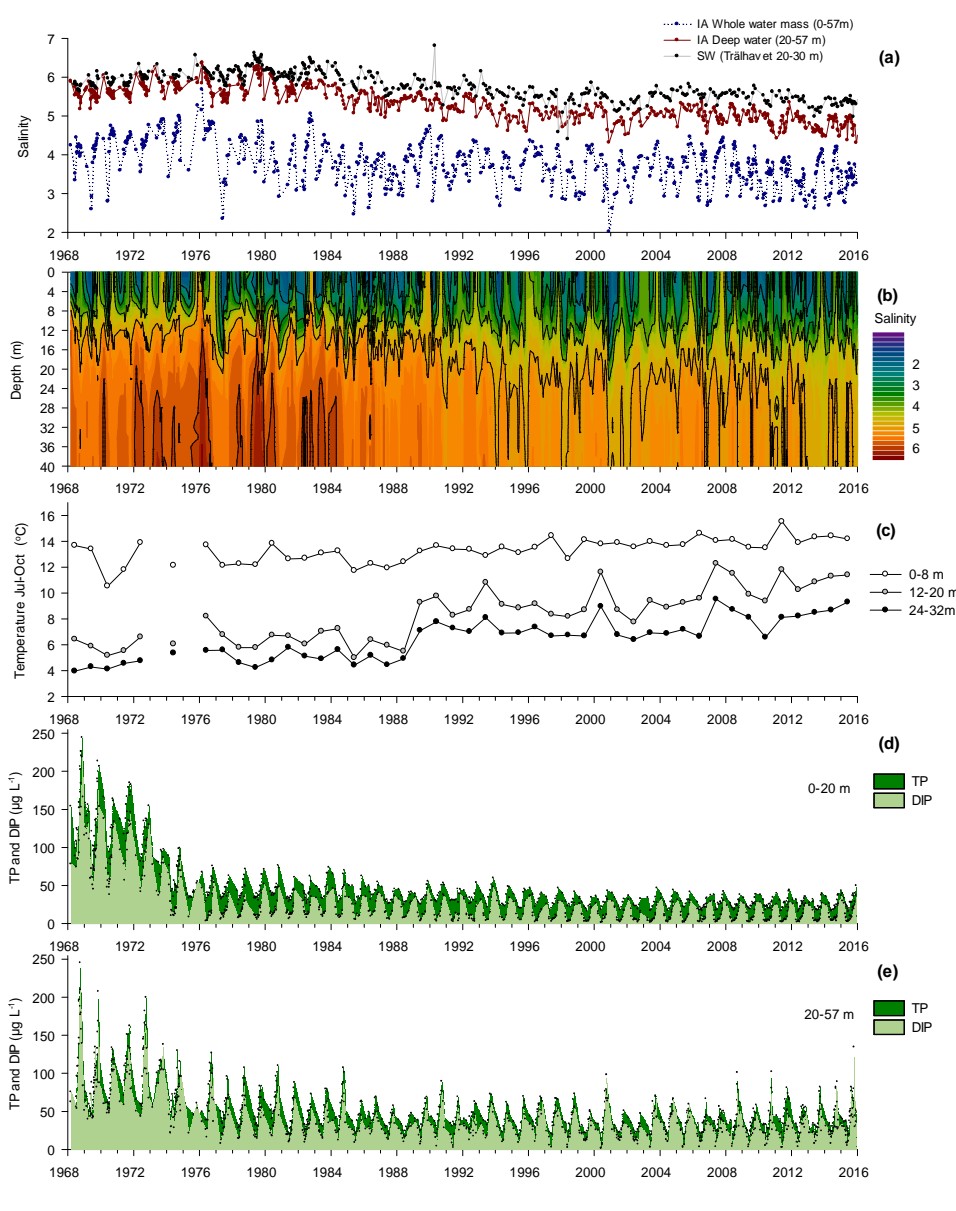

**Figure 3.**



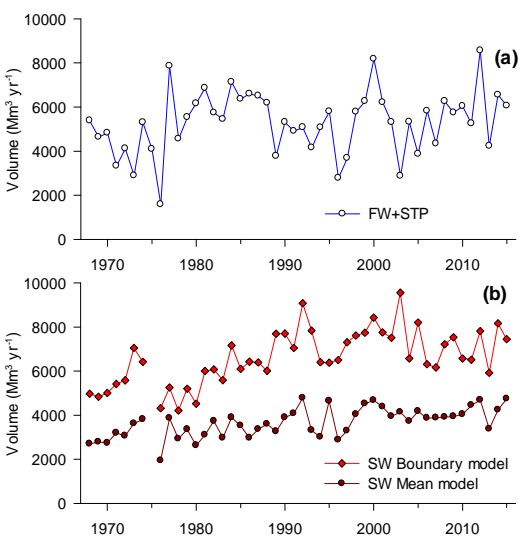

**Figure 4.**



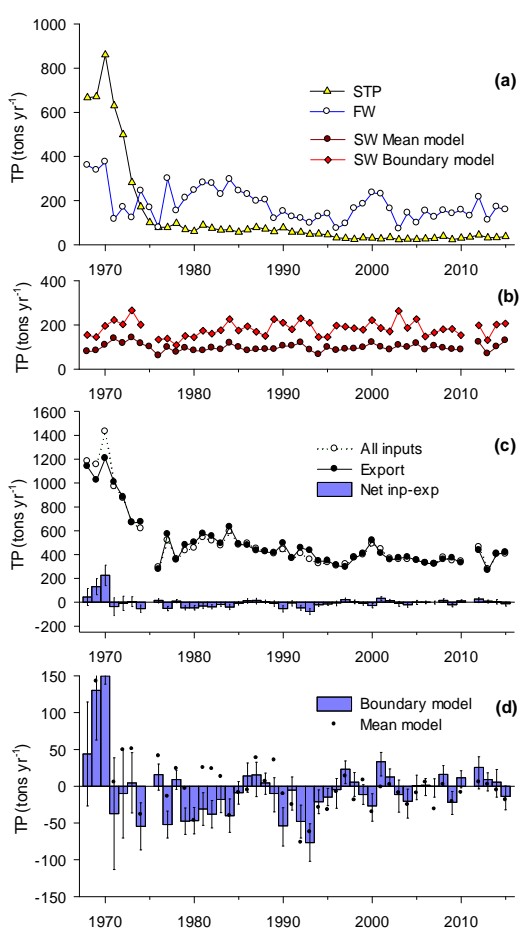

**Figure 5.**



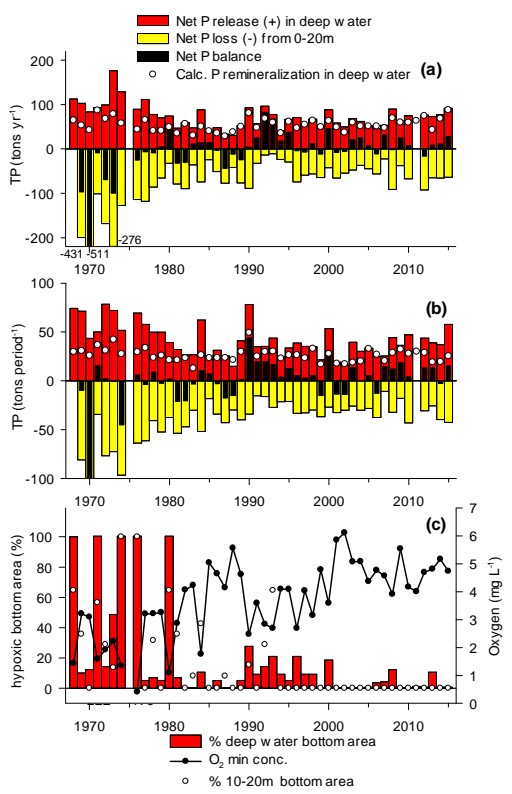

**Figure 6.**





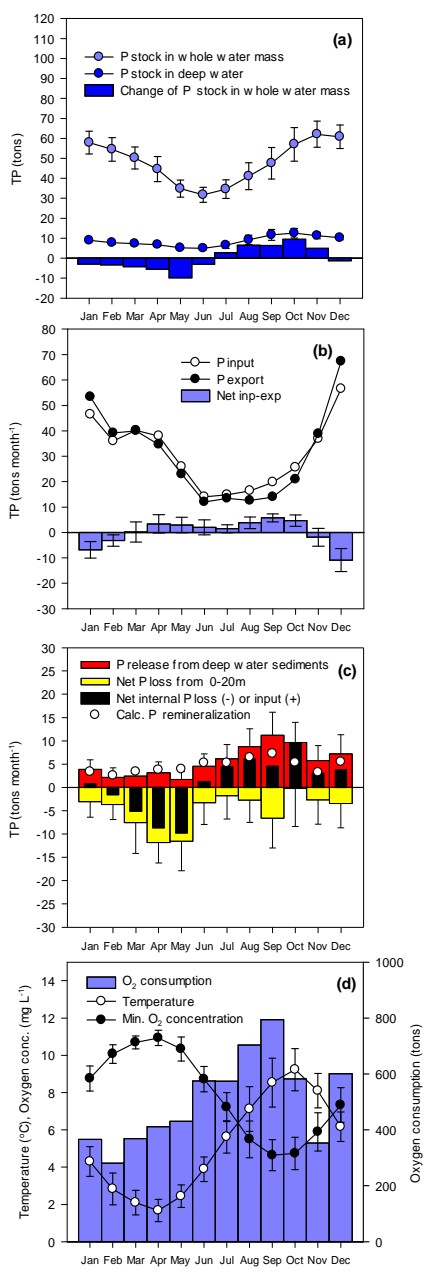

**Figure 7.**





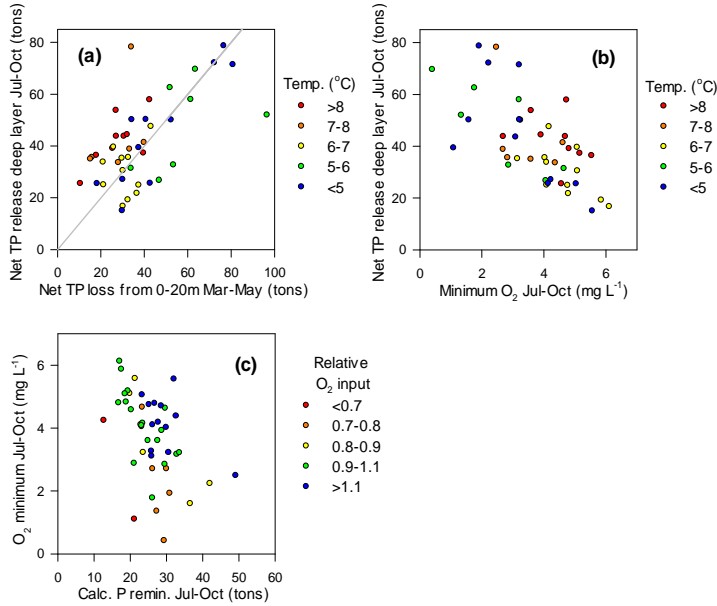

**Figure 8.**





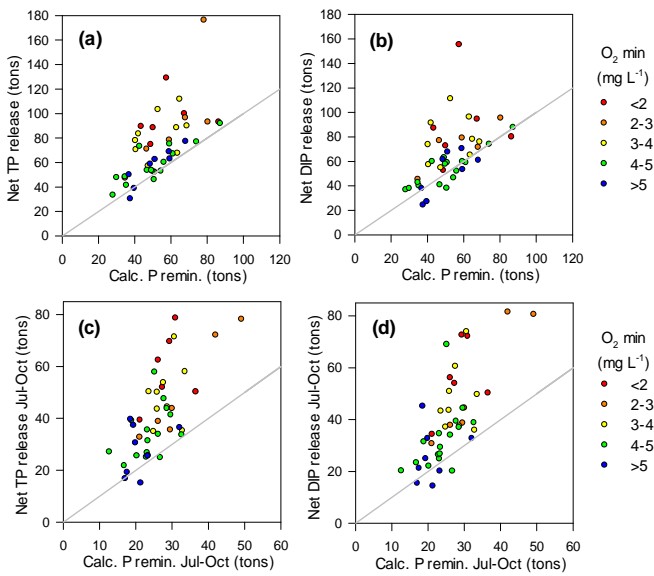

**Figure 9.**



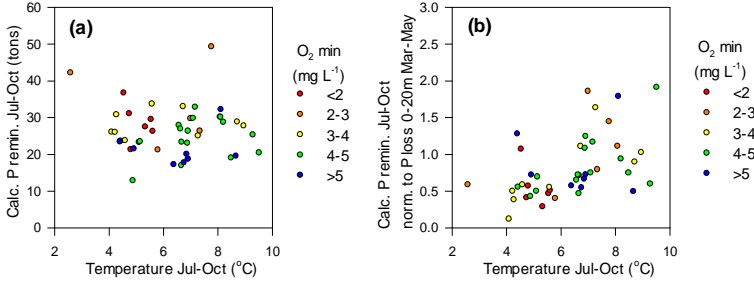

**Figure 10.**





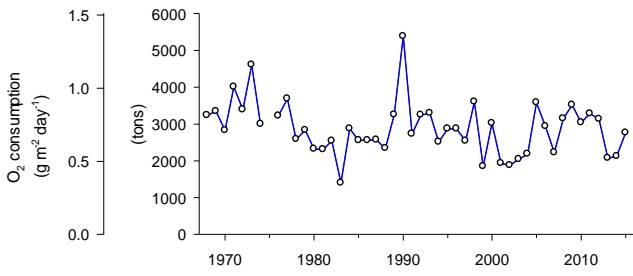

**Figure 11.**