# Peer review of "A Baltic Sea estuary as phosphorus source and sink after drastic load reduction: seasonal and long-term mass-balances for the Stockholm inner archipelago 1968-2015"

_Biogeosciences, 2017_

## Referee Comment (RC1) · Anonymous Referee #1 · 17 Jan 2018

Reviewer comments

Summary Walve et al. (bg-2017-496) present an interesting study where phosphorus dynamics in an estuary in the Baltic Sea are quantified using a box-model approach. The study period spans from 1968 to 2015, and has a seasonal temporal resolution. Box model budgets for water and salt are calculated from measured freshwater inflow and salinity from two stations. These water budgets are further used to infer phosphorus fluxes to and from the study estuary. In general, the study comprises of a straightforward model exercise where abundant monitoring data has been used to build the model. One of the main findings is the identification of the sediment as a sink of phosphorus in spring and as a source during summer and autumn. Overall, the manuscript is well written and easy to follow.

[Figure]

General comments 1) A clear set of scientific questions and/or hypotheses are not presented. This renders the manuscript to resemble more a data report or a technical note. Also, the point of view is on one particular system, which makes the study very local, as generalizations that could apply to other systems are not presented. The manuscript would benefit from a clear formulation of scientifically relevant research question and/or hypotheses, which would potentially be relevant to other systems as well. 2) As pointed out in the study, there are multiple previous studies focusing on the phosphorus dynamics in this particular system (e.g. Karlgren and Ljungström, 1975; Karlsson et al., 2010; Rydin et al. 2011; Almroth-Rosell et al. 2016). This means that the novelty aspects of this manuscript should be emphasized more. What are the main discrepancies compared to other studies, and why adding one more model study of this system is justified? This justification is even more important, as the key finding (sediment as a sink for P in spring and as a source in summer/autumn) is also rather well understood phenomenon in coastal systems overall.

Specific comments

Note: line numbers should be continuous throughout the manuscript, not start again at the beginning of each page.

Title The title is very concise, maybe even too much so. The seasonality behind the source and sink actions could be introduced in the title already.

Abstract P1, L10: In effect, two box models are used in the study, and the box models are vertically layered. This should be made clear here as well. P1, L10: Here and elsewhere: there is inconsistent use of terms "box-model" and "box model". Authors should choose either one and use the same term throughout. P1, L20: Replace stores with pools or storages. P1, L23: Sentence is incomplete, contributed to what?

Introduction P1, L31-32: Sentence is vague and does not read well, please rephrase. P3, L1: Remove "coastal", as estuaries are by definition coastal. If estuaries are to be emphasized here, then the sentence should be rephrased. In general, estuaries

have short residence time in the mixed layer, but is that the case in the Stockholm Archipelago? I would assume that the stratification and the relatively deep basins lead to high water residence times in the water masses close to the bottom, compared to the fast water exchange in the surface layer. P3, L6: "deteriorated badly" does not read well, rephrase. P3, L32: Is this not the case with all box models, that they are dynamic? Or is this a special case of a "dynamic" box model? If so, it should be elaborated here. P3, L34: Should be "focused". P4, L1-4: See above; here I would expect to see a set of research questions, objectives and/or hypotheses which would clarify the aim of the study and also make it easy to follow throughout the manuscript if the goals are met in the study.

Methods P4, L7: should be "comprises of". P4, L20: Does this mean that temperature was measured from inside the sampling vessels? Please clarify in text. P5, L8-13: Why four layers? What is the justification of this number of layers and the depth ranges of those layers? Please elaborate. P6, L20: Replace at with during.

Discussion P13, L21: Replace scale with resolution. P13, L21: The reader should be reminded at this point what is 1A. P13, L22: "...and has poor long-term P retention..." does not read well, rephrase. P13, L26-27: When both models yield similar results in P budgets, what is the motivation to use both? If the poorer prediction of water exchange in the mean model is not important (indicating decoupling between surface and bottom water masses), then why include the boundary model at all? P14, L22: Replace yearly with annual. P14, L18-19: This is an important factor causing bias in sediment studies; in practice, most sediment cores are taken from sedimentation areas (local bathymetric depressions) where sedimentation rates are higher than in their surrounding area in general. This matter should be elaborated here. P15, L10-11: What is the the significant difference between the inner and outer part here? Why do you think that the polychates do not have an effect there? This should be elaborated here. P15, L13-21: So where does the TP in the upper layer go? It seems like sedimentation is suggested, but it is vaguely formulated and should be made clearer. P16, L14: Iron

obviously plays a major role in the P cycling in the sediments, therefore the iron fluxes and concentrations presented on page 17 should be brought up already here. This would better justify the lengthy discussion about the role of Fe in the system. P17, L15: Is this mean annual concentration? Should be made clear. P17, L19-20: Due to salinity differences between the surface and the bottom layers in the system, this is more like a rule than an exception, is it not? Therefore the assumption of water masses being well-mixed seems not justified. P17, L32: Replace yearly with annual. P18, L10: Here, P is presumed to be deposited as organic P. Why iron-bound (inorganic) P is not considered here, as it clearly is a significant pathway for P sedimentation as presented before.

Conclusions Overall, this section is too long and would benefit from shortening. Ideally, conclusions are a concise presentation of the key findings, which relate to the research questions and/or hypotheses presented earlier. For instance, the seasonality aspects could be the main concluding remarks.

Tables Table 1: Replace yearly with annual in the caption Table 2: Replace yearly with annual in the caption

Figures Figure 1: Sampling points can not be distinguished well enough from their background, so they should be made larger and with better contrast. Figure 4: These time series seem to have an increasing trend over the study period. This could be emphasized by plotting a regression line of some kind to the plots. This would also allow the quantification of the annual increase in observed values. Figure 5: In panel (c), the net values are hard to read because of the scale issue. For that reason, the net values should be shown only in the separate panel (d). The motivation to show also the values from the mean model in panel (d) is unclear.

End of review.

---

## Referee Comment (RC2) · Anonymous Referee #2 · 18 Jan 2018

GENERAL COMMENTS: The manuscript by Walve et al. is a local study investigating whether the Stockholm inner Archipelago acts as a source or sink for phosphorus. They use a four-level box model based on observations to calculate the inflow and outflow of phosphorus to and from the study area as well as to calculate the retention of phosphorus within the area. The study is interesting and the manuscript is overall easy to follow.

SPECIFIC COMMENTS: The title is not specific enough, since it is a local study in a limited area and not valid in the entire Baltic Sea. The abstract is too long. It can be more concise. There are some text that would go under study site description and some text that is more of a discussion. The study is not put into a larger perspective, to what extent is this study interesting for the rest of the Europe or the World? Are there

any more studies performed in the same way, and what do they show? Weaknesses and strengths in in those and this study? There are other studies of the retention of phosphorus in the same area as this study, which are mentioned in the manuscript, but what is new with this study. What is the new scientific question? It is not necessary to use too many abbreviations since it makes the text unnecessarily complex to read. One example is the abbreviation of the water layers, D, M, LS, US. The Conclusion section is too long and should be more concise. It is too much of a discussion in it. The last section in the conclusion is not even discussed anywhere in the manuscript. From where was this concluded?

TECHNICAL CORRECTIONS: P1, L12: what do you mean with the "greater" Stockholm Area. However, it can be questioned if the information about the sewage treatment plants should be in the abstract at all. P1,L18: ". . .. bloom and is exported during winter. . .." exported to where? From the sediment to the water or to outer areas? P1, L22-L24 The word "probably" makes it sound as it is a discussion part and not anything that is concluded. P1, L32 would the word "occurrence" fit better than "incidence"? P3,L5 It is a bit unclear what you mean with small and medium towns. I would assume that it is a question of definition which can be different in different parts of the world. What do you mean? P3, L25 Actually this is discussed in the study by Almroth-Rosell et al. (2015), full reference below. P4, L13-L14 What is the reference for the description of the area? P4, L18 DIP is already defined above sections. P4, L29 STP is also already defined in above sections. P6, L16 Should it be "S in Fig. 1"? P7, L2 "LS" is not defined in the text. P7, L17 K as in K in fig. 1 ? or why have K within parenthesis? P7, L27 In PEXP-MM, what does MM mean? P7, L31 In PEXP-BM, BM is not defined as well. P8, L2 The net import should be defined once if it is the import-export or the opposite, but it is by definition a net result, and the word "net" should not be included as well when it is written out as P import-export. This goes for the entire manuscript. P8, L2 "closely mirrors", I am not sure that it can be seen so easy from the figure. What is the correlation coefficients? P9, L2 . . .. Higher in. . .. Higher than what? This can be seen also in other parts of the manuscript. Please go through and check

this. P10, L18 "closely mirrors", correlates? What is the correlation coefficients? P11, L1 6b, do you mean 6c? Correct also at other places throughout the manuscript. P11, L22 change July to August? P11, L23 insert water at the end or the row: surface water P11, L29-L30 "Although ...... among years." Can be seen where? P12, L19 insert "negatively" in front of "correlated". P14, L18-L31 I am not sure what you would like to say with this paragraph. Rewrite. P17, L1 Blomqvist et al. (2004) should be cited here, full reference below. P18, L26 "…. 1990 on..." Remove "on". P30, L18-L20 in c) and d) and e) it should be clarified that it is in the inner archipelago and not at a specific station. P30, L23-L25 Here it should be said something about the two models shown in fig. 4b. P30, L26 Replace the first comma with " of". P31, L14 Replace "yearly" with "annual" P31, L24 No P-value? P32, L1 Replace "yearly" with "annual"

Fig. 1 The letters showing the different stations cannot be seen properly. They are too small and the contrast with the background is too bad. Fig.3 It is hard to distinguish the dots in the legend from each other. Make larger, and change color is one suggestion.

REFERENCES: Almroth-Rosell, E., Eilola, K., Kuznetsov, I., Hall, P.O.J., Meier, H.E.M., 2015. A new approach to model oxygen dependent benthic phosphate fluxes in the Baltic Sea. J.Mar.Sys. 144, 127-141. Doi: 10.1016/j.jmarsys.2014.11.007

Blomqvist, S., Gunnars, A., Elmgren, R., 2004. Why the Limiting Nutrient Differs between Temperate Coastal Seas and Freshwater Lakes: A Matter of Salt. Limnol. Oceanogr. 49, 2236-2241.

---

## Author Comment (AC1) · 14 Feb 2018

Authors' response (AR) to comments by referee 1 (RC)

RC: "Walve et al. (bg-2017-496) present an interesting study where phosphorus dynamics in an estuary in the Baltic Sea are quantified using a box-model approach. The study period spans from 1968 to 2015, and has a seasonal temporal resolution. Box model budgets for water and salt are calculated from measured freshwater inflow and salinity from two stations. These water budgets are further used to infer phosphorus fluxes to and from the study estuary. In general, the study comprises of a straight-forward model exercise where abundant monitoring data has been used to build the model. One of the main findings is the identification of the sediment as a sink of phosphorus in spring and as a source during summer and autumn. Overall, the manuscript is well written and easy to follow." AR: We are pleased that our study was overall found interesting and easy to follow, and are thankful for the constructive comments. Although the finding of the sediment as seasonal source and sink is important, we consider the finding that seasonal P release is larger than legacy P release, and that legacy P release was clearly demonstrable only for a few initial years, as the main results. Moreover, seasonal P release nearly equals the sink meaning that annual net retention is very low.

RC: "The manuscript would benefit from a clear formulation of scientifically relevant research question and/or hypotheses, which would potentially be relevant to other systems as well." AR: We agree. As is also suggested in a comment below, we will add the requested text in the final paragraph of the introduction. Research questions include: 1. In an estuary that has long been subject to a very high P loading, is there a long-term influence of legacy P once the load has been drastically reduced? 2. What is the current annual P retention of the area? 3. To what extent does oxygen control seasonal P release and annual P retention? Can we expect improvements in oxygen conditions to increase long-term P retention? 4. What is the effect of the temperature increase from global warming on P release?

RC: "What are the main discrepancies compared to other studies, and why adding one more model study of this system is justified?" AR: This is the first temporally highly resolved dynamic box model study of the Stockholm IA that follows long-term P balance, including the period of change from high to low P load, and the first with seasonal resolution. Moreover, we link P retention to oxygen concentrations and oxygen consumption (from an oxygen budget). We find lower P retention than previous studies, even periods of net P export. We find that oxygen consumption from degradation of organic matter is predominantly directly linked to P release. We also find that low oxygen can promote an additional release of P. We will clarify this justification of our study in the final paragraph of the introduction. Discrepancies from other studies (e.g. in mean

annual P retention) are presently dealt with in the discussion, but will be clarified also in the introduction.

RC: "The title is very concise, maybe even too much so. The seasonality behind the source and sink actions could be introduced in the title already" AR: We suggest expanding the title to "A Baltic Sea estuary as phosphorus source and sink after drastic load reduction: seasonal and long-term mass-balances for the Stockholm inner archipelago 1968-2015". This emphasizes the long-term results and gives the exact location of the study.

RC: "Abstract P1, L10: In effect, two box models are used in the study, and the box models are vertically layered. This should be made clear here as well." AR: Agree, we will make it clear already here that we use two models, and that both are vertically layered.

RC: "P1, L10: Here and elsewhere: there is inconsistent use of terms "box-model" and "box model". Authors should choose either one and use the same term throughout." AR: We now consistently write "box model".

RC: "P1, L20: Replace stores with pools or storages." AR: Corrected.

RC: "P1, L23: Sentence is incomplete, contributed to what?" AR: Contributed to improved oxygen conditions. We have removed lines 21-24 that deal with reasons for oxygen increases in order to shorten and focus the abstract (as also suggested by reviewer 2). We replace them with a short sentence about temperature effect on oxygen consumption.

RC: "Introduction P1, L31-32: Sentence is vague and does not read well, please rephrase." AR: We agree, and will rephrase to "The occurrence of hypoxia is also influenced by variations in deep-water ventilation, which depend on basin morphology and meteorological forcing (Zhang et al., 2010)."

RC: "P3, L1: Remove "coastal", as estuaries are by definition coastal. If estuaries are

to be emphasized here, then the sentence should be rephrased." AR: Corrected and rephrased.

RC: "In general, estuaries have short residence time in the mixed layer, but is that the case in the Stockholm Archipelago? I would assume that the stratification and the relatively deep basins lead to high water residence times in the water masses close to the bottom, compared to the fast water exchange in the surface layer." AR: The water mass in the upper 10 meters is relatively large, so the difference is not substantial. The fresh-water residence time is in the order of weeks to months. We add a reference to Engqvist and Andrejev (2003), who give estimates of water retention times. This reference is cited elsewhere in the manuscript and is in the reference list.

RC: "P3, L6: "deteriorated badly" does not read well, rephrase." AR: Changed to: . . . deteriorated seriously . . .

RC: "P3, L32: Is this not the case with all box models, that they are dynamic? Or is this a special case of a "dynamic" box model? If so, it should be elaborated here." AR: We are referring to the relatively high temporal resolution and the imbalanced instantaneous salt inflow and outflow, i.e. it is not just based on annual means assuming steady-state. We rephrase to clarify this.

RC: "P4, L1-4: See above; here I would expect to see a set of research questions, objectives and/or hypotheses which would clarify the aim of the study and also make it easy to follow throughout the manuscript if the goals are met in the study." AR: We agree. We will add the requested text. See response to second comment above.

RC: "Methods P4, L7: should be "comprises of"." AR: We do not agree, and neither does our dictionary.

RC: "P4, L20: Does this mean that temperature was measured from inside the sampling vessels? Please clarify in text." AR: Thermometer inside water sampler, then in-situ with CTD. Will be clarified.

RC: "P5, L8-13: Why four layers? What is the justification of this number of layers and the depth ranges of those layers? Please elaborate." AR: We used a box model with fixed depth layers and need three layers to separate deep water (primarily influenced by inflowing salt water) from a pycnocline depth layer and surface water. We further divided the surface layer since at high fresh water flows there was occasionally a shallower halocline. This is elaborated on page 6, line 20-24. We move these lines to the first paragraph in section 2.3 and add explanation of other layers.

RC: "P6, L20: Replace at with during." AR: Corrected.

RC: "Discussion P13, L21: Replace scale with resolution." AR: Changed.

RC: "P13, L21: The reader should be reminded at this point what is IA." AR: Agree. Corrected.

RC: "P13, L22: "…and has poor long-term P retention.."does not read well, rephrase." AR: Rephrased. "Poor" is changed to "low".

RC: "P13, L26-27: When both models yield similar results in P budgets, what is the motivation to use both? If the poorer prediction of water exchange in the mean model is not important (indicating decoupling between surface and bottom water masses), then why include the boundary model at all?" AR: The boundary model should be more correct if looking at the P balance of the inner area. However, the mean model is more representative regarding the internal dynamics, based on central stations data. We think we have to present all the results to be clear. The similarity of the results of both models is in itself an interesting result that adds credence to the analysis.

RC: "P14, L22: Replace yearly with annual." AR: Corrected.

RC: "P14, L18-19: This is an important factor causing bias in sediment studies; in practice, most sediment cores are taken from sedimentation areas (local bathymetric depressions) where sedimentation rates are higher than in their surrounding area in general. This matter should be elaborated here." AR: We elaborate in lines 20-25, but

will add also this aspect. We rephrase this part and the rest of this paragraph to make it clearer (in response also to comment by reviewer 2).

RC: "P15, L10-11: What is the significant difference between the inner and outer part here? Why do you think that the polychates do not have an effect there? This should be elaborated here." AR: Other factors than the polychaetes can explain the observations in the cited paper. The inner part of the IA has a complex water circulation. We conclude we cannot see an effect, but lack a solid basis for further speculation.

RC: "P15, L13-21: So where does the TP in the upper layer go? It seems like sedimentation is suggested, but it is vaguely formulated and should be made clearer." AR: Agree, we modify the second sentence.

RC: "P16, L14: Iron obviously plays a major role in the P cycling in the sediments, therefore the iron fluxes and concentrations presented on page 17 should be brought up already here. This would better justify the lengthy discussion about the role of Fe in the system." AR: We moved the sentence on Fe loading to the area to page 16.

RC: "P17, L15: Is this mean annual concentration? Should be made clear." AR: Corrected.

RC: "P17, L19-20: Due to salinity differences between the surface and the bottom layers in the system, this is more like a rule than an exception, is it not? Therefore the assumption of water masses being well-mixed seems not justified." AR: There is hardly any horizontal salinity gradient in the deep water, so this is likely not a problem. The vertical salinity differences are handled by the model.

RC: "P17, L32: Replace yearly with annual." AR: Corrected.

RC: "P18, L10: Here, P is presumed to be deposited as organic P. Why iron-bound (inorganic) P is not considered here, as it clearly is a significant pathway for P sedimentation as presented before." AR: The intended point here is that the organic P load was very likely higher in the 1970s than in later years, which should contribute to high

oxygen consumption, and mostly low O2 concentrations, unless O2 inputs are particularly large. (There is probably significant inorganic P sedimentation too, but this will not contribute to oxygen consumption.) We rephrase to clarify.

RC: "Conclusions Overall, this section is too long and would benefit from shortening. Ideally, conclusions are a concise presentation of the key findings, which relate to the research questions and/or hypotheses presented earlier. For instance, the seasonality aspects could be the main concluding remarks." AR: Yes, we suggest moving most of the last two paragraphs to new section 4.6.

RC: "Tables Table 1: Replace yearly with annual in the caption Table 2: Replace yearly with annual in the caption" AR: Corrected.

RC: "Figure 1: Sampling points can not be distinguished well enough from their background, so they should be made larger and with better contrast." AR: Corrected.

RC: "Figure 4: These time series seem to have an increasing trend over the study period. This could be emphasized by plotting a regression line of some kind to the plots. This would also allow the quantification of the annual increase in observed values." AR: In principle yes. But then we should for consistency add such lines in many of the following graphs too. Many changes are not steady trends but rather short term fluctuations and thus a regression line could be a bit misleading. We prefer mentioning it in the text.

RC: "Figure 5: In panel (c), the net values are hard to read because of the scale issue. For that reason, the net values should be shown only in the separate panel (d). The motivation to show also the values from the mean model in panel (d) is unclear." AR: We agree net values in panel (c) can be removed. We think results from the mean model are important for comparative reasons. However, we have results of the mean model in Table 3 and can remove it in fig. 5d.

---

## Author Comment (AC2) · 14 Feb 2018

Authors' response (AR) to comments by referee 2 (RC)

RC: "The manuscript by Walve et al. is a local study investigating whether the Stockholm inner Archipelago acts as a source or sink for phosphorus. They use a four-level box model based on observations to calculate the inflow and outflow of phosphorus to and from the study area as well as to calculate the retention of phosphorus within the area. The study is interesting and the manuscript is overall easy to follow." AR: We are pleased that our study was overall found interesting and the manuscript easy to follow, and are thankful for the constructive comments.

RC: "The title is not specific enough, since it is a local study in a limited area and

not valid in the entire Baltic Sea." AR: We suggest expanding the title to "A Baltic Sea estuary as phosphorus source and sink after drastic load reduction: seasonal and long-term mass-balances for the Stockholm inner archipelago 1968-2015"

RC: "The abstract is too long. It can be more concise. There are some text that would go under study site description and some text that is more of a discussion." AR: We agree it can be shortened. We suggest deleting the sentence starting in the end of row 11 and the following sentence. We modify the sentence starting on row 13. The "discussion" sentences (lines 21-24) about causes of oxygen improvements will be replaced by a single sentence about oxygen consumption, P release and temperature, something like "Increasing temperatures, stimulating deep-water oxygen consumption rates in early summer, have counteracted the effects of lowered organic matter sedimentation on oxygen conditions"

RC: "The study is not put into a larger perspective, to what extent is this study interesting for the rest of the Europe or the World?" AR: We will add sentences in the introduction of the general function of estuaries as filter for nutrients. We also modify the final part of the introduction to clarify the motivation for the study (see response below) and stress the importance of studies in the Stockholm inner archipelago, which has often been used as an example in the debate about effects of P and N mitigation to coastal waters (Boesch et al. 2006, Schindler and Vallentyne 2008, Schindler et al. 2008). The latter reference is new and will be added to the reference list (Schindler et al.: Eutrophication of lakes cannot be controlled by reducing nitrogen input: Results of a 37-year whole-ecosystem experiment. PNAS 105: 11254-11258, 2008). Our study also contributes to clarifying the importance of internal seasonal P cycling versus legacy P loads, and that oxygen control of P release alone is an oversimplification. We have already mentioned these problems generally in the introduction, but will modify conclusions to better link to general interest.

RC: "Are there any more studies performed in the same way, and what do they show? Weaknesses and strengths in in those and this study?" AR: We will add more general

discussion of results of some other studies, e.g. the already cited Testa and Kemp (2008) and Staehr et al. (2017). Weaknesses and strengths of previous studies in the same area are already discussed in section 4.1, but we will modify the final paragraph of the introduction to bring up the differences already here.

RC: "There are other studies of the retention of phosphorus in the same area as this study, which are mentioned in the manuscript, but what is new with this study. What is the new scientific question?" AR: This is the first temporally highly resolved dynamic box model study of the study area that follows the long-term P balance, including the period of change from high to low P load, and the first with seasonal resolution. Moreover, we link P retention to oxygen concentrations and oxygen consumption (from oxygen budget). We find lower P retention than previous studies, even periods of net P export. We find that oxygen consumption from degradation of organic matter is predominantly directly linked to P release. We also find that low oxygen can promote an additional release of P. We will clarify this in the final paragraph of the introduction. Discrepancies from other studies (mean annual P retention) are taken up in the discussion. We will also add text in the final paragraph of the introduction clarifying the scientific questions of the study. Such questions include: 1. In an estuary that has long been subject to a very high P loading, is there a long-term influence of legacy P once the load has been drastically reduced? 2. What is the current annual P retention of the area? 3. To what extent does oxygen control seasonal P release and annual P retention? Can we expect improvements in oxygen conditions to increase long-term P retention? 4. What is the effect of the temperature increase from global warming on P release?

RC: "It is not necessary to use too many abbreviations since it makes the text unnecessarily complex to read. One example is the abbreviation of the water layers, D, M, LS, US." AR: We will avoid using the water layer abbreviations in the text and write "deep water " etc, including in methods P7 L1-3. However, we still need to introduce these abbreviations for use in the equations. We will remove the abbreviations introduced on

P7 L23-31.

RC: "The Conclusion section is too long and should be more concise. It is too much of a discussion in it. The last section in the conclusion is not even discussed anywhere in the manuscript. From where was this concluded?" AR: We suggest moving most of the last two paragraphs of "Conclusions" to a new final section 4.6. We will rewrite other parts of the conclusion to make it more concise.

RC: "P1, L12. what do you mean with the "greater" Stockholm Area. However, it can be questioned if the information about the sewage treatment plants should be in the abstract at all. " AR: We agree and will delete this sentence.

RC: "P1,L18. ". . .. bloom and is exported during winter. . ..." exported to where? From the sediment to the water or to outer areas?" AR: To outer areas. Rephrased.

RC: "P1, L22-L24 The word "probably" makes it sound as it is a discussion part and not anything that is concluded." AR: We suggest removing the section about reasons for improved oxygen conditions from the abstract (lines 21-24), since this is not a main topic of the paper and needs more studies.

RC: "P1, L32 would the word "occurrence" fit better than "incidence"?" AR: We agree and will rephrase this sentence to make it clearer.

RC: "P3,L5 It is a bit unclear what you mean with small and medium towns. I would assume that it is a question of definition which can be different in different parts of the world. What do you mean?" AR: Largest "medium" towns are up to 150 000 inhabitants. Clarified.

RC: " P3, L25 Actually this is discussed in the study by Almroth-Rosell et al. (2015), full reference below." AR: Yes, this refers to the open Baltic Sea but we add this reference.

RC: "P4, L13-L14 What is the reference for the description of the area?" AR: The reference is the SMHI basin register (SMHI, 2003), used elsewhere. This reference is now added also here.

RC: " P4, L18 DIP is already defined above sections." AR: Corrected.

RC: " P4, L29 STP is also already defined in above sections. " AR: Corrected.

RC: " P6, L16 Should it be "S in Fig. 1"? " AR: Yes, clarified.

RC: "P7, L2 "LS" is not defined in the text. " AR: Defined on P5 L10. However, we will remove layer abbreviations from the running text, keeping them only for the equations.

RC: "P7, L17 K as in K in fig. 1 ? or why have K within parenthesis? " AR: Yes, clarified.

RC: "P7, L27 In PEXP-MM, what does MM mean? " AR: Mean model. The abbreviation is removed here, and defined and used in table only.

RC: " P7, L31 In PEXP-BM, BM is not defined as well. " AR: Boundary model. The abbreviation is removed here, and defined and used in tables only.

RC: "P8, L2 The net import should be defined once if it is the import-export or the opposite, but it is by definition a net result, and the word "net" should not be included as well when it is written out as P import-export. This goes for the entire manuscript." AR: Yes, corrected.

RC: " P8, L2 "closely mirrors", I am not sure that it can be seen so easy from the figure. What is the correlation coefficients? " AR: This is not a straightforward calculation since stations are often not sampled the same day. Using dates that are the same day or very close we find a linear regression coefficient $r2$ of 0.75. The mirroring will also be affected by delays, which are obscuring direct regression fits. However, we will remove the word "closely".

RC: "P9, L2 . . .. Higher in. . ... Higher than what? This can be seen also in other parts of the manuscript. Please go through and check this. " AR: Here higher after the shift 1989 than in the previous period. Clarified.

RC: "P10, L18 "closely mirrors", correlates? What is the correlation coefficients?" AR: $r2=0.97$. We add this number in the text.

RC: " P11, L1 6b, do you mean 6c? Correct also at other places throughout the manuscript." AR: Yes, it should be 6c (6c should also be 6b in other places).

RC: "P11, L22 change July to August? " AR: We focus here on the July-October period.

RC: "P11, L23 insert water at the end or the row: surface water" AR: Modified.

RC: " P11, L29-L30 "Although . . .. . . among years." Can be seen where? " AR: Variability is indicated in fig 7c but is not shown explicitly for each year. We add this figure reference.

RC: "P12, L19 insert "negatively" in front of "correlated". " AR: Modified.

RC: "P14, L18-L31 I am not sure what you would like to say with this paragraph. Rewrite." AR: We rephrase this paragraph and also divide it into two. The first deals with the reasons why there is an apparent discrepancy between accumulation of sediments and low P retention (sometimes even net P release). The reasons are the problem of defining representative P burial from sediment accumulation rates and legacy P release from organic matter degradation. The second paragraph continues with further aspects on legacy P release.

RC: " P17, L1 Blomqvist et al. (2004) should be cited here, full reference below." AR: Agree. We add this reference here too.

RC: " P18, L26 ". . .. 1990 on. . ." Remove "on"." AR: Corrected.

RC: " P30, L18-L20 in c) and d) and e) it should be clarified that it is in the inner archipelago and not at a specific station." AR: Will be clarified in (d) and (e).

RC: " P30, L23-L25 Here it should be said something about the two models shown in fig. 4b." AR: Yes, added.

RC: " P30, L26 Replace the first comma with " of". " AR: Corrected.

RC: "P31, L14 Replace "yearly" with "annual" " AR: Corrected.

RC: "P31, L24 No P-value? " AR: As stated in Results (P13 L5-6), there is reason to be careful here due to interdependencies of concentrations and model results. This is why no p-value was added. Disregarding this, there is a significant linear regression, y=6.31-0.10x, r2=0.23, p<0.001 for all data. (y=6.68-0.10x, r2=0.38, p<0.0001 if normalized O2 input values <0.8 are excluded). We suggest adding p-value for the first regression within parenthesis with reference to results.

RC: "P32, L1 Replace "yearly" with "annual" " AR: Corrected.

RC: "Fig. 1 The letters showing the different stations cannot be seen properly. They are too small and the contrast with the background is too bad." AR: OK, will be modified.

RC: " Fig.3 It is hard to distinguish the dots in the legend from each other. Make larger, and change color is one suggestion. " AR: OK, will be modified.

---

## Author Response (AR1)

**Authors' response (AR) to comments by referee 1 (RC)**

RC: "Walve et al. (bg-2017-496) present an interesting study where phosphorus dynamics in an estuary in the Baltic Sea are quantified using a box-model approach. The study period spans from 1968 to 2015, and has a seasonal temporal resolution. Box model budgets for water and salt are calculated from measured freshwater inflow and salinity from two stations. These water budgets are further used to infer phosphorus fluxes to and from the study estuary. In general, the study comprises of a straightforward model exercise where abundant monitoring data has been used to build the model. One of the main findings is the identification of the sediment as a sink of phosphorus in spring and as a source during summer and autumn. Overall, the manuscript is well written and easy to follow."

AR: We are pleased that our study was found interesting and easy to follow, and are grateful for the constructive comments. Although the finding of the sediment as seasonal source and sink is important, we consider the finding that legacy P release was clearly demonstrable only for the initial 10-20 years, and that later on seasonal P release is larger than any possible legacy P release, as the main results. Moreover, seasonal P release nearly equals the sink meaning that present annual net P retention is very low.

RC: "The manuscript would benefit from a clear formulation of scientifically relevant research question and/or hypotheses, which would potentially be relevant to other systems as well."

AR: We agree. We have added the requested text in the final paragraph of the introduction. Research questions are added to include:

1. In an estuary that has long been subject to a very high P loading, is there a long-term influence of legacy P once the load has been drastically reduced? What is the current annual P retention of the area?

2. To what extent does oxygen control seasonal P release and annual P retention? Can we expect improvements in oxygen conditions to increase long-term P retention?

3. Can an effect of the temperature increase on oxygen consumption and P release be identified?

**RC: "What are the main discrepancies compared to other studies, and why adding one more model study of this system is justified?"**

AR: This is the first temporally highly resolved dynamic box model study of the Stockholm IA that follows the long-term P balance, including the period of change from high to low P load, and the first with seasonal resolution. Moreover, we link P retention to oxygen concentrations and oxygen consumption (from a deep-water oxygen budget). We find lower P retention than previous studies, even periods of net P export. We find that oxygen consumption from degradation of organic matter is predominantly directly linked to P release. We also find that low oxygen can promote an additional release of P. We have clarified this justification of our study in the final paragraph of the introduction. Discrepancies from other studies (e.g. in mean annual P retention) are presently dealt with in the discussion, but are now noted also in the introduction.

RC: "The title is very concise, maybe even too much so. The seasonality behind the source and sink actions could be introduced in the title already"

AR: We have expanded the title to "A Baltic Sea estuary as phosphorus source and sink after drastic load reduction: seasonal and long-term mass-balances for the Stockholm inner archipelago 1968-2015". This emphasizes the long-term results, the seasonality aspect, and gives the exact location of the study.

RC: "Abstract P1, L10: In effect, two box models are used in the study, and the box models are vertically layered. This should be made clear here as well."

AR: We now make it clear already in the Abstract that we use more than one model, and that both are vertically layered.

RC: "P1, L10: Here and elsewhere: there is inconsistent use of terms "box-model" and "box model". Authors should choose either one and use the same term throughout."

AR: We now consistently write "box model".

RC: "P1, L20: Replace stores with pools or storages."

AR: Corrected.

**RC: "P1, L23: Sentence is incomplete, contributed to what?"**

AR: Contributed to improved oxygen conditions. We have removed lines 21-24 that dealt with reasons for oxygen increases in order to shorten and focus the abstract (as also suggested by reviewer 2). We have replaced them with a short sentence about temperature effect on oxygen consumption.

RC: "Introduction P1, L31-32: Sentence is vague and does not read well, please rephrase."

AR: We agree, and have rephrased.

RC: "P3, L1: Remove "coastal", as estuaries are by definition coastal. If estuaries are to be emphasized here, then the sentence should be rephrased."

AR: Corrected and rephrased.

RC: "In general, estuaries have short residence time in the mixed layer, but is that the case in the Stockholm Archipelago? I would assume that the stratification and the relatively deep basins lead to high water residence times in the water masses close to the bottom, compared to the fast water exchange in the surface layer."

AR: due to the estuarine circulation and the relatively large water mass in the upper 10 meters, the retention time of the deep-water is shorter than the surface water. We have added a sentence in section 2.1 with reference to Engqvist and Andrejev (2003), who give estimates of water retention times. This reference is cited elsewhere in the manuscript and was already in the reference list.

RC: "P3, L6: "deteriorated badly" does not read well, rephrase."

AR: Changed to: ... deteriorated seriously ...

RC: "P3, L32: Is this not the case with all box models, that they are dynamic? Or is this a special case of a "dynamic" box model? If so, it should be elaborated here."

AR: We intended to stress the relatively high temporal resolution and the imbalanced instantaneous salt inflow and outflow, i.e. that the model is not just based on annual means assuming steady-state. We have rephrased to clarify this.

RC: "P4, L1-4: See above; here I would expect to see a set of research questions, objectives and/or hypotheses which would clarify the aim of the study and also make it easy to follow throughout the manuscript if the goals are met in the study."

AR: We agree. We will add the requested text. See response to second comment above.

RC: "Methods P4, L7: should be "comprises of"."

AR: We do not agree, and neither does our dictionary.

RC: "P4, L20: Does this mean that temperature was measured from inside the sampling vessels? Please clarify in text."

AR: Thermometer inside water sampler, then in-situ with CTD. Clarified.

RC: "P5, L8-13: Why four layers? What is the justification of this number of layers and the depth ranges of those layers? Please elaborate."

AR: We used a box model with fixed depth layers and need three layers to separate deep water (primarily influenced by inflowing salt water) from a pycnocline depth layer and surface water. We further divided the surface layer since at high fresh water flows there was occasionally a shallower halocline. This was elaborated on page 6, line 20-24. We have moved these lines to the first paragraph in section 2.3 and added further explanation.

RC: "P6, L20: Replace at with during."

AR: Corrected. This sentence is moved to first paragraph of section 2.3.

RC: "Discussion P13, L21: Replace scale with resolution."

AR: Changed.

RC: "P13, L21: The reader should be reminded at this point what is IA."

AR: Agree. Corrected.

RC: "P13, L22: "...and has poor long-term P retention.." does not read well, rephrase."

AR: Rephrased.

RC: "P13, L26-27: When both models yield similar results in P budgets, what is the motivation to use both? If the poorer prediction of water exchange in the mean model is not important (indicating decoupling between surface and bottom water masses), then why include the boundary model at all?"

AR: The boundary model should be more correct if looking at the P balance of the inner area. However, the mean model is more representative regarding the internal dynamics, based on central stations data. We think we have to present all the results to be clear. The similarity of the results of both models is in itself an interesting result that adds credence to the analysis. We added clarification in this paragraph.

RC: "P14, L22: Replace yearly with annual."

AR: Corrected. Replaced throughout the manuscript.

RC: "P14, L18-19: This is an important factor causing bias in sediment studies; in practice, most sediment cores are taken from sedimentation areas (local bathymetric depressions) where sedimentation rates are higher than in their surrounding area in general. This matter should be elaborated here."

AR: We have elaborated and clarified in this paragraph (in response also to comment by reviewer 2). The final sentences of this paragraph is now a separate paragraph.

RC: "P15, L10-11: What is the significant difference between the inner and outer part here? Why do you think that the polychates do not have an effect there? This should be elaborated here."

AR: Factors other than the polychaetes could explain the observations in the cited paper. The inner part of the IA has a complex water circulation. We conclude we cannot see an effect, but lack a solid basis for further speculation.

RC: "P15, L13-21: So where does the TP in the upper layer go? It seems like sedimentation is suggested, but it is vaguely formulated and should be made clearer."

AR: We agree, and have modified the second sentence.

RC: "P16, L14: Iron obviously plays a major role in the P cycling in the sediments, therefore the iron fluxes and concentrations presented on page 17 should be brought up already here. This would better justify the lengthy discussion about the role of Fe in the system."

AR: We now bring this up earlier, and have moved the sentence on Fe loading to second paragraph in section 4.3.

RC: "P17, L15: Is this mean annual concentration? Should be made clear."

AR: Corrected.

RC: "P17, L19-20: Due to salinity differences between the surface and the bottom layers in the system, this is more like a rule than an exception, is it not? Therefore the assumption of water masses being well-mixed seems not justified."

AR: There is hardly any horizontal salinity gradient in the deep water, so this is likely not a problem. The vertical salinity differences are handled by the model.

RC: "P17, L32: Replace yearly with annual."

AR: Corrected.

RC: "P18, L10: Here, P is presumed to be deposited as organic P. Why iron-bound (inorganic) P is not considered here, as it clearly is a significant pathway for P sedimentation as presented before."

AR: The intended point here is that the organic P load was very likely higher in the 1970s than in later years, which should contribute to high oxygen consumption, and mostly low O2 concentrations, unless O2 inputs are particularly large. There is probably significant inorganic P sedimentation too, but this will not contribute to oxygen consumption. We have rephrased to clarify.

RC: "Conclusions Overall, this section is too long and would benefit from shortening. Ideally, conclusions are a concise presentation of the key findings, which relate to the research questions and/or hypotheses presented earlier. For instance, the seasonality aspects could be the main concluding remarks."

AR: Yes, we have now shortened the conclusions by moving most of the last two paragraphs to the new section 4.6.

RC: "Tables Table 1: Replace yearly with annual in the caption Table 2: Replace yearly with annual in the caption"

AR: Corrected.

RC: "Figure 1: Sampling points can not be distinguished well enough from their background, so they should be made larger and with better contrast."

**AR: Corrected.**

RC: "Figure 4: These time series seem to have an increasing trend over the study period. This could be emphasized by plotting a regression line of some kind to the plots. This would also allow the quantification of the annual increase in observed values."

AR: In principle yes. But then we should for consistency add such lines in many of the following graphs too. Many changes are not steady trends but rather short term fluctuations and thus a regression line could be a bit misleading. We prefer mentioning it in the text.

RC: "Figure 5: In panel (c), the net values are hard to read because of the scale issue. For that reason, the net values should be shown only in the separate panel (d). The motivation to show also the values from the mean model in panel (d) is unclear."

AR: We agree that the net values in panel (c) can be removed. The results of the mean model are important for comparisons, but since they are reported in Table 3, they can be removed from fig. 5d.

**Authors' response (AR) to comments by referee 2 (RC)**

RC: "The manuscript by Walve et al. is a local study investigating whether the Stockholm inner Archipelago acts as a source or sink for phosphorus. They use a four-level box model based on observations to calculate the inflow and outflow of phosphorus to and from the study area as well as to calculate the retention of phosphorus within the area. The study is interesting and the manuscript is overall easy to follow."

AR: We are pleased that our study was found interesting and the manuscript easy to follow, and are grateful for the constructive comments.

RC: "The title is not specific enough, since it is a local study in a limited area and not valid in the entire Baltic Sea."

AR: We have expanded the title to "A Baltic Sea estuary as phosphorus source and sink after drastic load reduction: seasonal and long-term mass-balances for the Stockholm inner archipelago 1968-2015"

RC: "The abstract is too long. It can be more concise. There are some text that would go under study site description and some text that is more of a discussion."

AR: We agree to some shortening. We deleted the sentence starting in the end of row 11 and the following sentence. We modified the sentence starting on row 13. The "discussion" sentences (lines 21-24) about causes of oxygen improvements have been replaced by a single sentence about oxygen consumption, P release and temperature.

RC: "The study is not put into a larger perspective, to what extent is this study interesting for the rest of the Europe or the World?"

AR: We have added perspective in the introduction concerning the general importance this type of studies, citing reviews by Duarte et al. (2015), Testa et al. (2017), Regnier et al. (2013), and Cloern et al. (2014), to better link to general interest and stress the importance of this type of study. We have modified the final part of the introduction to clarify the motivation for the study (see response below) and added text to stress the importance of studies in the Stockholm inner archipelago, which has often been used as an example in the debate about effects of P and N mitigation to coastal waters (Boesch et al. 2006, Schindler and Vallentyne 2008, Schindler et al. 2008). The latter reference is new and has been added to the reference list.

Our study also contributes to clarifying the importance of internal seasonal P cycling versus legacy P loads, and that oxygen control of P release alone is an oversimplification. We have also added a management point in the new section 4.6 about artificial oxygenation of bottom waters.

**RC: "Are there any more studies performed in the same way, and what do they show? Weaknesses and strengths in in those and this study?"**

AR: We have added more general introduction and further discussion of results of some other studies, e.g. the already cited Testa and Kemp (2008) and Staehr et al. (2017), and the new reference Boynton et al. (2018). Weaknesses and strengths of previous studies in the same area are already discussed in section 4.1, but we have modified the final paragraph of the introduction to bring up the differences already here.

**RC: "There are other studies of the retention of phosphorus in the same area as this study, which are mentioned in the manuscript, but what is new with this study. What is the new scientific question?"**

AR: This is the first temporally highly resolved dynamic box model study of the study area that follows the long-term P balance, including the period of change from high to low P load, and the first with seasonal resolution. Moreover, we link P retention to oxygen concentrations and oxygen consumption (from oxygen budget). We find lower P retention than previous studies, even periods of net P export. We find that oxygen consumption from degradation of organic matter is predominantly directly linked to P release. We also find that low oxygen can promote an additional release of P. We

have clarified this in the final paragraph of the introduction. Discrepancies from other studies (mean annual P retention) are taken up in the discussion. We have also added text in the final paragraph of the introduction clarifying the scientific questions of the study. Such questions include:

1. In an estuary that has long been subject to a very high P loading, is there a long-term influence of legacy P once the load has been drastically reduced? What is the current annual P retention of the area?

2. To what extent does oxygen control seasonal P release and annual P retention? Can we expect improvements in oxygen conditions to increase long-term P retention?

3. Can an effect of the temperature increase on oxygen consumption and P release be identified?

RC: "It is not necessary to use too many abbreviations since it makes the text unnecessarily complex to read. One example is the abbreviation of the water layers, D, M, LS, US."

AR: We have removed most of the water layer abbreviations from the text and write "deep water " etc, including in methods P7 L1-3. However, we still need to introduce these abbreviations for use in the equations. We have removed the abbreviations introduced on P7 L23-31.

RC: "The Conclusion section is too long and should be more concise. It is too much of a discussion in it. The last section in the conclusion is not even discussed anywhere in the manuscript. From where was this concluded?"

AR: We have moved most of the last two paragraphs of "Conclusions" to a new final section 4.6 and have made other parts of the conclusion more concise.

RC: "P1, L12. what do you mean with the "greater" Stockholm Area. However, it can be questioned if the information about the sewage treatment plants should be in the abstract at all. "

AR: We agree and have deleted this sentence.

RC: "P1,L18. ".... bloom and is exported during winter...." exported to where? From the sediment to the water or to outer areas?"

AR: To outer areas. Rephrased.

RC: "P1, L22-L24 The word "probably" makes it sound as it is a discussion part and not anything that is concluded."

AR: We have removed the section about reasons for improved oxygen conditions from the abstract (lines 21-24), since this is not a main topic of the paper and needs more studies.

RC: "P1, L32 would the word "occurrence" fit better than "incidence"?"

AR: We agree and have rephrased this sentence to make it clearer.

RC: "P3,L5 It is a bit unclear what you mean with small and medium towns. I would assume that it is a question of definition which can be different in different parts of the world. What do you mean?"

AR: Largest "medium" towns are up to 150 000 inhabitants. Clarified.

RC: "P3, L25 Actually this is discussed in the study by Almroth-Rosell et al. (2015), full reference below."

AR: We now discuss and have added this reference.

RC: "P4, L13-L14 What is the reference for the description of the area?"

AR: The reference is the SMHI basin register (SMHI, 2003), used elsewhere. This reference is now added also here.

RC: "P4, L18 DIP is already defined above sections."

AR: Corrected.

RC: "P4, L29 STP is also already defined in above sections."

AR: Corrected.

RC: "P6, L16 Should it be "S in Fig. 1"?"

AR: Yes, clarified.

RC: "P7, L2 "LS" is not defined in the text. "

AR: Defined on P5 L10. However, we have remove layer abbreviations from the running text, keeping them only for the equations.

RC: "P7, L17 K as in K in fig. 1 ? or why have K within parenthesis? "

AR: Yes, modified.

RC: "P7, L27 In PEXP-MM, what does MM mean?"

AR: Mean model. The abbreviation is removed here, and defined and used in table only.

RC: "P7, L31 In PEXP-BM, BM is not defined as well. "

AR: Boundary model. The abbreviation is removed here, and defined and used in tables only.

RC: "P8, L2 The net import should be defined once if it is the import-export or the opposite, but it is by definition a net result, and the word "net" should not be included as well when it is written out as P import-export. This goes for the entire manuscript."

AR: Yes, corrected.

RC: "P8, L2 "closely mirrors", I am not sure that it can be seen so easy from the figure. What is the correlation coefficients? "

AR: This is not a straightforward calculation since stations are often not sampled the same day. Using dates that coincide or are very close we find a linear regression coefficient  $r^2$  of 0.75. The mirroring may be visible also when there are delays, but delays obscure direct regression fits. However, we removed the word "closely".

RC: "P9, L2 . . . . Higher in. . ... Higher than what? This can be seen also in other parts of the manuscript. Please go through and check this. "

AR: Here higher after the shift 1989 than in the previous period. Clarified here and elsewhere.

RC: "P10, L18 "closely mirrors", correlates? What is the correlation coefficients?"

AR:  $r^2$ =0.97. We add this number in the text.

RC: "P11, L1 6b, do you mean 6c? Correct also at other places throughout the manuscript."

AR: Yes, it should be 6c (6c should also be 6b in other places).

RC: "P11, L22 change July to August?"

AR: We focus here on the July-October period.

RC: "P11, L23 insert water at the end or the row: surface water"

AR: Modified.

RC: "P11, L29-L30 "Although . . . . . among years." Can be seen where? "

AR: Variability is indicated in fig 7c but is not shown explicitly for each year. We added this figure reference.

RC: "P12, L19 insert "negatively" in front of "correlated". "

AR: Modified.

RC: "P14, L18-L31 I am not sure what you would like to say with this paragraph. Rewrite."

AR: We have rephrased this paragraph and divided it into two. The first deals with the reasons why there is an apparent discrepancy between accumulation of sediments and low P retention (sometimes even net P release). The reasons are the problem of defining representative P burial from sediment accumulation rates and legacy P release from organic matter degradation. The second paragraph continues with further aspects on legacy P release.

RC: "P17, L1 Blomqvist et al. (2004) should be cited here, full reference below."

AR: Agree. We add this reference here too.

RC: " P18, L26 ". . . . 1990 on. . ." Remove "on"."

AR: Corrected.

RC: "P30, L18-L20 in c) and d) and e) it should be clarified that it is in the inner archipelago and not at a specific station."

AR: Now clarified in (d) and (e).

RC: "P30, L23-L25 Here it should be said something about the two models shown in fig. 4b."

AR: Yes, added.

RC: "P30, L26 Replace the first comma with " of". "

AR: We have corrected text here by removing "water flows of".

RC: "P31, L14 Replace "yearly" with "annual" " AR: Corrected.

RC: "P31, L24 No P-value? "

AR: As stated in Results (P13 L5-6), there is reason to be careful here due to interdependencies of concentrations and model results. This is why no p-value was added. Disregarding this, there is a significant linear regression, y=6.31-0.10x,  $r^2$ =0.23, p<0.001 for all data. (y=6.68-0.10x,  $r^2$ =0.38, p<0.0001 if normalized O2 input values <0.8 are excluded). We have added p-value for the first regression within parenthesis with reference to results section.

RC: "P32, L1 Replace "yearly" with "annual" "

AR: Corrected.

RC: "Fig. 1 The letters showing the different stations cannot be seen properly. They are too small and the contrast with the background is too bad."

AR: OK, modified.

RC: "Fig.3 It is hard to distinguish the dots in the legend from each other. Make larger, and change color is one suggestion. "

AR: In Fig. 3a symbols have been enlarged and also have changed order to correspond to order of lines in graph. Symbols in Fig 3c are also enlarged.

[revised manuscript text omitted]

Theoretically expected The P release rates (P02) theoretically expected from mineralization of organic matter were calculated from deep-water O2 consumption, assuming mineralization of organic matter of Redfield elemental composition (see methods). In Mmost years, the annual P02 calculated from organic matter decomposition was close to the modelled annual TP and DIP release rates (Fig. 6a-b), but tended to be lower than, Mmodelled TP and DIP release tended to be higher than P02-in years with low minimum O2 concentrations (e.g. 1984, 1992-93, 1996, 2000, 2008 and 2012) (Fig. 6a and Fig. 9a-b). When the comparison was restricted to July-October, P02 explained less of the P release (Fig. 6be and Fig. 9-c-d), with

30 largest deviation for September (Fig. 7). Conversely, the modelled P release was generally somewhat lower than  $P_{O2}$  in May-June (Fig. 7, Suppl. 3).

**3.9 Factors affecting oxygen conditions**

The observed  $O_2$  minimum correlates with modelled  $O_2$  consumption in July-October (Fig. 8c, here converted to P release, see above). Since the calculation of  $O_2$  consumption depends on the observed  $O_2$  minimum concentration, a correlation is expected and should be interpreted with caution. We find a similar correlation between the  $O_2$  minimum, which is usually in

5 September, and the O2 consumption in July (data not shown). Although our model results indicate that the O2 consumption rate influences the O2 concentrations of the deep water, hypoxia usually-primarily developed in years with low O2 
[revised manuscript text omitted]

y=-2.69+1.70x,  $r^2=0.46$ , p<0.0001 for all values,  $r^2=10.6+0.91x$ ,  $r^2=0.20$ , p<0.05 for O2 minimums >4; **d:** y=-8.55+1.89x,  $r^2=0.48$ , p<0.0001 for all values, y=6.27+1.04x,  $r^2=0.20$ , p<0.05 for O2 minimums >4).

Figure 10. (a) Yearly Annual oxygen consumption July-October 1969-2015 (1975 and 2011 excluded) recalculated to
theoretical deep-water P remineralization, as a function of temperature. The data set is colour-coded according to minimum oxygen concentration July-October: (y=28.3-0.31x, r2=0.006, p=0.60).

(b) Aes in (a) but with theoretical P remineralization is-normalized to P loss from 0-20 m layer in March-May- (y=-0.030+0.13x, r2=0.25, p<0.001).

10 **Figure 11.** July-October oOxygen consumption July October-in the Stockholm inner archipelago deep\_-water 1968-2015, calculated with the mean model.